# Turning Adaptation into Assets: Cross-Domain Bridging for Online Vision-Language Navigation

**Zixuan Hu** [1]  **Xuantuo Huang** [2]  **Yancheng Li** [2]  **Yichun Hu** [1]  **Shengyong Xu** [2]  **Ling-Yu Duan** [† 1 3]

## Abstract

Navigating under non-stationary environment shifts poses a critical challenge for a Vision-and-Language Navigation (VLN) agent deployed in the wild. Yet, existing Test-Time Adaptation (TTA) methods for VLN largely treat online adaptation as transient, isolated updates, leading to catastrophic forgetting and negative transfer. To overcome these issues, we propose **I**nter-**D**omain Bridg**E** with Historical **A**ssets (**IDEA**), a novel TTA framework that transforms adaptation into the accumulation and composition of assets. Specifically, IDEA introduces soft prompts optimized via a Fisher-guided weighting scheme to capture the transferable knowledge. These optimized prompts are then augmented with domain coordinates to form a dynamic asset library. Leveraging this library, IDEA constructs a cross-domain bridge by projecting the target domain onto the convex hull of historical knowledge. These designs form a complementary loop: the evolving library underpins bridge construction, while the bridge provides superior initialization to accelerate asset optimization. Extensive experiments across REVERIE, R2R, and R2R-CE benchmarks demonstrate the consistent superiority of IDEA, showcasing its ability to enable training-free adaptation via asset sharing.

## 1. Introduction

Vision-and-Language Navigation (VLN) serves as a fundamental embodied task, enabling the agent to ground language instructions into an action sequence that leads to the goal location (Anderson et al., 2018; Gu et al., 2022;

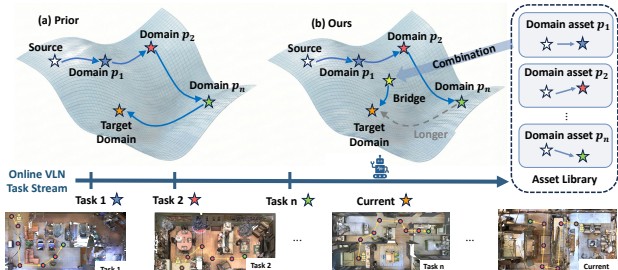

*Figure 1.* Illustration of different adaptation formulations in VLN. (a) Prior works: A series of isolated domain transfer tasks. (b) Ours: Turns adaptation into accumulation and reuse of composable assets.

Song et al., 2025). In real-world navigation, it rarely resembles the curated conditions of training, as agents inevitably encounter unseen environments where visual appearances and spatial layouts can differ substantially across episodes. Moreover, VLN exhibits rapid intra-episode shifts, with observations changing markedly along the trajectory. Such dynamic distribution shifts cause significant performance drops (Gu et al., 2022; Guhur et al., 2021), posing a key challenge for reliable deployment in the wild.

To mitigate distribution shifts with minimal overhead, Test-Time Adaptation (TTA) has emerged as a critical paradigm, enabling models to adapt to target distributions via online updates (Iwasawa & Matsuo, 2021; Liang et al., 2023; Alfarra et al., 2024). In embodied navigation, existing TTA methods broadly fall into two categories. *Uncertainty self-training* leverages entropy measures to drive step-wise updates for improved action selection (Gao et al., 2024). *Feedback-driven adaptation* incorporates corrective signals from foundation models (FMs) or human feedback (Kim et al., 2025); however, the high inference cost of FMs renders such guidance impractical for real-time deployment. Given frequent scene transitions in VLN, both categories rely on continual adaptation to stay aligned with the evolving input stream.

Despite their advantages, current studies model online VLN as a series of isolated domain-transfer tasks (see Fig. 1(a)), ignoring correlations across recurring or related contexts. This yields two critical bottlenecks: 1) *Catastrophic forgetting.* Online updates on a fixed parameter set often overwrite earlier adaptations, hindering the ability to leverage historical experience upon revisiting scenes. 2) *Negative transfer.*

[1]School of Computer Science, Peking University, Beijing, China [2]School of Electronics, Peking University, Beijing, China [3]Peng Cheng Laboratory, Shenzhen, China. Correspondence to: Ling-Yu Duan <lingyu@pku.edu.cn>.

*Proceedings of the 43rd International Conference on Machine Learning*, Seoul, South Korea. PMLR 306, 2026. Copyright 2026 by the author(s).

Without modeling domain relations, updates derived from one context may be blindly applied to a dissimilar one, introducing mismatched priors and degrading performance. Such issues contribute to wasted adaptation effort and impede building transferable knowledge across vast environments.

To address these challenges, we propose a novel TTA framework "**I**nter-**D**omain Bridg**E** with Historical **A**ssets (**IDEA**)", which reformulates VLN adaptation as the accumulation and composition of knowledge across correlated domains, as shown in Fig. 1(b). IDEA adopts a *knowledge-as-asset* view: instead of treating past training as transient or domain-specific, it progressively distills the historical knowledge into a lightweight and context-aware asset library. This structured repository directly addresses forgetting by preserving history, while allowing assets to be selectively retrieved and recomposed to prevent negative transfer. Leveraging this library, IDEA identifies a training-free shortcut to the target domain, constructing an optimal bridge that effectively bypasses the lengthy adaptation trajectory.

Specifically, IDEA introduces compact prompts to encode adaptation knowledge, optimizing them to reduce the source–target gap in the representation space. To enhance knowledge transferability, we propose a Fisher-guided weighting scheme to adaptively prioritize the policy-sensitive parameters, isolating task-essential priors from domain-specific noise. These optimized prompts are then enriched with domain coordinates to form reuse-ready assets. Building upon the library, we search the closest target projection onto the convex hull spanned by historical assets under Wasserstein distance, producing a cross-domain bridge that shortens the adaptation pathway. To keep this step lightweight, we derive a closed-form solution via the KKT conditions, avoiding iterative solvers. These two designs operate as a complementary loop: as the asset library evolves, it offers richer building blocks for bridge construction, and the bridge in turn provides a better initialization that accelerates subsequent asset optimization.

We evaluate the effectiveness and generalizability of our method through experiments on the discrete VLN benchmarks REVERIE (Qi et al., 2020) and R2R (Anderson et al., 2018), achieving state-of-the-art results with average improvements of **+2.5%** SR and **+1.9%** SPL. Furthermore, we showcase its superior performance in addressing shifts in continuous benchmark R2R-CE (Krantz et al., 2020), yielding consistent gains compared to existing methods.

**Contributions.** 1) To the best of our knowledge, we are the first to transform adaptation knowledge into structured assets, pioneering a plug-and-play reuse paradigm for Test-Time Adaptation in VLN. 2) We propose a novel TTA framework IDEA, which introduces Fisher-guided prompt tuning for transferable knowledge encoding, and constructs a projection-based bridge to provide a training-free shortcut

to the target domain. 3) We provide a theoretical analysis demonstrating that our bridging mechanism stably reduces the generalization risk on unseen target domains. 4) Extensive experiments across four models and three benchmarks demonstrate IDEA's superiority. Furthermore, we validate the portability of asset library, showing its potential for scalable sharing to bypass the cold-start phase of new agents.

## 2. Related Work

### 2.1. Vision-Language Navigation

Vision-Language Navigation (VLN) requires an embodied agent to navigate to a target location in a 3D environment based on language instructions and visual observations (Gu et al., 2022; Song et al., 2025). In terms of architecture, early approaches formulated VLN as a sequential decision-making problem, employing Recurrent Neural Networks (RNNs) to map sensory inputs to actions (Fried et al., 2018; Hong et al., 2020). Subsequently, the advent of multimodal pre-training with Transformers (Vaswani et al., 2017) established a dominant paradigm, leveraging large-scale pre-training to encode robust cross-modal alignment (Hao et al., 2020; Chen et al., 2021; Hong et al., 2021; Huo et al., 2023; Zhang et al., 2025). To optimize these architectures, existing methods primarily rely on offline Imitation Learning to clone expert behaviors (An et al., 2023; Liu et al., 2024c; Zheng et al., 2025). Many works also incorporate reinforcement learning to refine policies beyond fixed supervised trajectories (Chen et al., 2022a; Gao et al., 2025; Xu et al., 2025). Despite these advancements, current methods typically operate under a static "Train-then-Deploy" paradigm, neglecting the utilization of test-time data. The ability of an agent to accumulate knowledge during the testing process would greatly enhance its practical value.

### 2.2. Test-Time Adaptation

Test-Time Adaptation aims to enhance the performance on out-of-distribution samples during inference (Liang et al., 2023; Lim et al., 2023; Lee et al., 2023). Existing TTA methods generally rely on uncertainty minimization (Niu et al., 2023; Hu et al., 2025b;c), batch normalization calibration (Mirza et al., 2022; Zhao et al., 2023a; Tan et al., 2025), or auxiliary self-supervised task (*e.g.* masked reconstruction (Mirza et al., 2023)) to provide effective proxy signals for reducing domain gaps (Boudiaf et al., 2022; Liu et al., 2024a). In the navigation context, exploration primarily falls into two groups: uncertainty minimization and feedback-driven adaptation. The former focuses on enforcing consistency constraints across temporal sequences to mitigate uncertainty (Gao et al., 2024). The latter leverages foundation models or human interaction as guidance to provide reward signals for reinforcement learning (Kim et al., 2025; Ko et al., 2025). However, these approaches treat adaptation as isolated tasks, failing to accumulate reusable knowledge for efficient, long-term generalization.

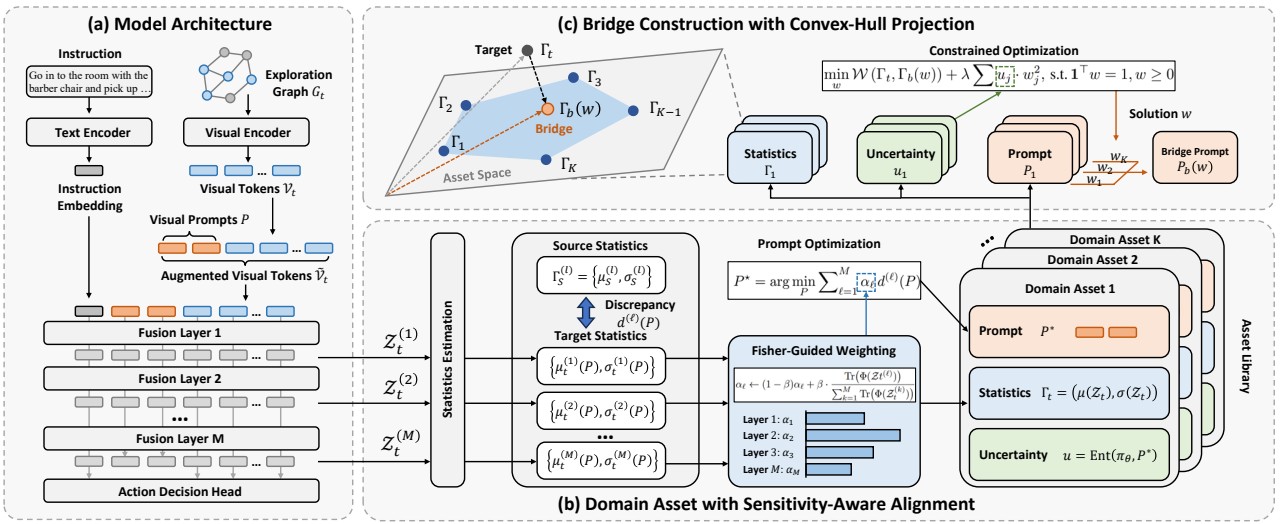

*Figure 2.* Overview of our IDEA method. (*a*) Meta-framework of the VLN models, where dual encoders extract multi-modal tokens, followed by a fusion transformer and a decision head. (*b*) Through our-derived Fisher-guided weighting term, IDEA optimizes soft prompts for sensitivity-aware alignment across fusion layers, forming triplet-structured assets (Sec. 4.1). (*c*) Leveraging the asset library, IDEA identifies an efficient adaptation shortcut by computing the optimal projection onto the convex hull spanned by historical assets (Sec. 4.2).

## 3. Preliminary

**Task Definition.** We consider the Vision-and-Language Navigation (VLN) task in a streaming test-time setting. The agent is provided with a pre-trained policy $\pi_\theta$, parameterized by $\theta$. During testing, the agent receives a sequence of navigation tasks $\mathcal{X} = \{X_1, \dots, X_N\}$. Each task $X_i = (I_i, s_0)$ consists of a natural language instruction $I_i$ and a visual observation of a 360° panoramic view at the initial point $s_0$. Starting from $s_0$, the agent iteratively selects an action until a stop is chosen, producing a trajectory:

$$\tau_i = \{(s_t, a_t)\}_{t=0}^{T_i-1}, \quad a_t \sim \pi_\theta(\cdot \mid s_t, I_i), \qquad (1)$$

where $T_i$ denotes total number of steps for task $X_i$.

**Model Architecture.** During navigation, the policy model progressively constructs an undirected exploration graph $G_t = (V_t, E_t)$, where $V_t$ denotes navigable nodes and $E_t$ encodes their connectivity. It processes the current state with a dual-branch encoder: a visual branch extracts node-wise tokens $\mathcal{V}_t \in \mathbb{R}^{N \times C}$ from $G_t$ using a pre-trained Vision Transformer, while a language branch encodes the instruction $I$ into a fixed embedding $\mathcal{I} \in \mathbb{R}^C$.

The model then fuses the visual tokens $\mathcal{V}_t$ with the instruction embedding $\mathcal{I}$ using a multi-layer transformer, yielding a set of multi-modal representations:

$$\mathcal{Z}_t = \phi(\mathcal{V}_t, \mathcal{I}), \quad \mathcal{Z}_t = \{z_i\}_{i=1}^N \in \mathbb{R}^{N \times C}, \qquad (2)$$

where $\phi(\cdot)$ denotes the fusion function, $z_i$ denotes the fused feature for the $i$-th candidate node, $N$ denotes the number of candidate nodes, and $C$ is the feature dimension. Finally, a decision head maps each $z_i$ to a scalar score and outputs the next action by choosing the highest-scoring candidate.

## 4. Methodology

In this paper, we propose IDEA (**I**nter-**D**omain Bridg**E** with Historical **A**ssets), a novel TTA framework for VLN. IDEA incorporates two core designs: Domain asset library with sensitivity-aware alignment (Sec. 4.1), and Bridge construction mechanism via convex-hull projection (Sec. 4.2). Together, these two components synergize to facilitate efficient, robust adaptation and continuous knowledge reuse. An overview of the framework is illustrated in Fig. 2.

### 4.1. Turning Adaptation into Assets

Existing VLN adaptation methods typically cast optimization as a series of isolated transfer tasks, resulting in overwritten knowledge and neglecting inter-domain connections. To break this isolation, we reformulate adaptation as the continuous accumulation of portable assets, moving from domain-specific updates to decoupled knowledge storage. Ideally, these assets should be *plug-and-play*, *transferable*, and *retrievable*, allowing for seamless integration and reuse. To realize this, we construct a prompt-based representation equipped with sensitivity-aware alignment, serving as the building block for our asset library.

**Soft Visual Prompts.** To enable a plug-and-play mechanism, we represent each knowledge as $L$ learnable prompt tokens $P := \{p_i\}_{i=1}^L$ with $p_i \in \mathbb{R}^C$. At navigation step $t$, the visual encoder outputs node-wise visual tokens $\mathcal{V}_t \in \mathbb{R}^{N \times C}$. We inject the soft prompt into the visual tokens by appending it to the token sequence, yielding:

$$\tilde{\mathcal{V}}_t = [P; \mathcal{V}_t] \in \mathbb{R}^{(L+N) \times C}. \qquad (3)$$

These augmented tokens $\tilde{\mathcal{V}}_t$ are then fed into the cross-modal fusion module together with the instruction embedding. In this way, the prompt can guide visual-language alignment without altering the frozen policy parameters.

**Multi-Layer Alignment.** To construct a reusable knowledge asset, a learned prompt must effectively bridge the distribution gap between the target environment and the source-trained backbone. Leveraging the source latent space as a robust anchor, we introduce a hierarchical alignment mechanism that regularizes prompt-augmented representations to match source-domain statistics. By decomposing the cross-modal $\phi$ into $M$ layers, the knowledge asset captures structural distribution shifts at different levels of abstraction.

Formally, we precompute the source feature statistics (mean and standard deviation), denoted as $\Gamma_S^{(\ell)} = \{\mu_S^{(\ell)}, \sigma_S^{(\ell)}\}$ for $\ell \in \{1, \ldots, M\}$, using a small subset of source data (128 samples). At each navigation step $t$, after injecting $P$, we extract the fused tokens $\mathcal{Z}_t^{(\ell)}(P)$ from layer $\ell$ and estimate their online statistics $\Gamma_t^{(\ell)}(P)$ over navigable nodes. The prompt is optimized to minimize the distributional divergence via the following moment-matching loss:

$$d^{(\ell)}(P) = \|\mu_S^{(\ell)} - \mu_t^{(\ell)}(P)\|_2 + \|\sigma_S^{(\ell)} - \sigma_t^{(\ell)}(P)\|_2. \quad (4)$$

where $\|\cdot\|_2$ denotes $\mathcal{L}_2$ norm. To enable adaptive alignment, the final objective is a weighted sum, with $\alpha_\ell$ controlling the contribution of each layer to the asset construction:

$$P^\star = \arg\min_P \sum_{\ell=1}^M \alpha_\ell\, d^{(\ell)}(P). \quad (5)$$

**Fisher-Guided Weighting.** To enhance asset transferability, we constrain the prompts to capture transferable task priors while suppressing overfitting to domain-specific factors. Our intuition relies on distinguishing between spurious and functional alignment: if a layer's optimization contributes solely to statistical matching but fails to influence the policy decisions, it implies the adaptation is merely fitting irrelevant distributional noise (Lee et al., 2024; Hu et al., 2024). In contrast, dynamics that significantly impact the output reflect the encoding of task-essential semantics. Therefore, we analyze the sensitivity of each layer's representation relative to the decision output to guide the weighting process.

In principle, such sensitivity is measured by the curvature of the policy objective, *i.e.*, the Hessian matrix. This second-order quantity describes how the policy output changes in response to perturbations in given parameters. However, computing the Hessian is computationally prohibitive for online inference, as evident in the following decomposition:

$$H(z) = \nabla_z^2(-\log \pi(a \mid z)) = -\nabla_z \left( \frac{\nabla_z \pi(a \mid z)}{\pi(a \mid z)} \right)$$
$$= -\frac{\pi(a \mid z)\nabla_z^2 \pi(a \mid z) - \nabla_z \pi(a \mid z)\nabla_z \pi(a \mid z)^\top}{\pi(a \mid z)^2},$$
$$(6)$$

where $H(z)$ denotes the Hessian matrix of representation $z$. To alleviate the computational burden, we derive a tractable proxy for the Hessian using first-order gradients. Since the ground-truth action is unavailable at test time, we reformulate the loss by treating the model's prediction as a soft label and taking the expectation with respect to the policy distribution. Under this framework, the expected Hessian simplifies to the Fisher Information Matrix, which requires only a single gradient computation:

$$\Phi(z) := \mathbb{E}_{a \sim \pi_\theta(\cdot)}\left[ \nabla_z \log \pi_\theta(a)\, \nabla_z \log \pi_\theta(a)^\top \right]. \quad (7)$$

We instantiate $z$ as the fused tokens $\mathcal{Z}^{(\ell)}$ at layer $\ell$, computed without prompt injection. For the Fisher matrix, we utilize its trace as a scalar proxy for layer-wise sensitivity. The normalized trace is then used to update the weight $\alpha_\ell$ in Eq. 5 via an exponential moving average:

$$\alpha_\ell \leftarrow (1 - \beta)\alpha_\ell + \beta \cdot \frac{\mathrm{Tr}\big(\Phi(\mathcal{Z}t^{(\ell)})\big)}{\sum_{k=1}^M \mathrm{Tr}\big(\Phi(\mathcal{Z}_t^{(k)})\big)}, \quad (8)$$

where $\beta$ is the smoothing coefficient, set to 0.1 by default.

**Triplet-Structured Asset.** Based on the above mechanism, we formalize adaptation knowledge not merely as optimized parameters, but as structured assets indexed within the domain manifold. To ensure retrieval and reuse, we augment the learned prompt with domain coordinates and quality metrics. Formally, each asset is defined as a triplet $\mathcal{A} := \{P^*, \Gamma, u\}$, comprising three complementary components: 1) *a learned soft prompt* $P^*$ that encodes task-relevant adaptation knowledge and directly modulates the policy; 2) *feature statistics* $\Gamma_t = \{\mu(\mathcal{Z}_t), \sigma(\mathcal{Z}_t)\}$ extracted from the final fusion layer without prompt injection, which serve as a prompt-independent descriptor of the environment; and 3) *an uncertainty score* $u = Ent(\pi_{\theta, P^*})$ defined as the prediction entropy at current step with $P^*$, which reflects the reliability of the adapted behavior with $\mathcal{A}$.

### 4.2. Cross-Domain Bridge Construction

Building upon accumulated assets, reusing prior domain knowledge offers an effective shortcut for bridging the gap between the source domain and the test-time target. A natural baseline is to perform hard retrieval, selecting the nearest-neighbor asset based on statistical similarity. However, a novel environment often exhibits partial overlap with multiple domains, rendering retrieval brittle and prone to mismatch. To fully exploit the representational power of the library, we cast bridge construction as finding the closest projection on the convex hull of assets, as shown in Fig. 2(c).

**Convex-Hull Projection Fitting.** Given the asset library $\mathcal{M} = \{\mathcal{A}_i\}_{i=1}^K$, we introduce mixture weights $w \in \mathbb{R}^K$ to formulate the projection as a soft composition. Specifically, we employ shared coefficients to linearly interpolate in both the parameter space (prompts) and the statistical space (domain coordinates). This unified projection yields an adaptation bridge in the form of a composite prompt, along with an induced distribution:

$$P_b(w) = \sum_{j=1}^K w_j \cdot P_j, \ \Gamma_b(w) = \sum_{j=1}^K w_j \cdot \Gamma_j, \quad (9)$$

where the subscript $b$ denotes the bridge.

To determine the optimal mixture weights, we project the target statistics $\Gamma_t$ onto $\Gamma_b(w)$ by minimizing the Wasserstein distance, assuming the feature statistics follow multivariate Gaussian distributions. To enhance robustness, we incorporate an uncertainty-aware regularizer that suppresses large weights on unreliable assets via their uncertainty scores $u_j$. The resulting constrained optimization problem is given by:

$$\min_w \mathcal{W}\left(\Gamma_t, \Gamma_b(w)\right) + \lambda \sum u_j \cdot w_j^2, \text{ s.t. } \mathbf{1}^\top w = 1, w \geq 0, \tag{10}$$

where $\mathcal{W}(\cdot, \cdot)$ denotes the 2-Wasserstein distance between the Gaussian distributions parameterized by the respective statistics, and $\lambda$ controls the strength of the regularization.

**Closed-Form Solution.** Directly optimizing Eq. (10) via gradient descent incurs substantial training overhead. Instead, we exploit the quadratic structure of the Wasserstein alignment objective and derive a closed-form solution via the Karush–Kuhn–Tucker (KKT) conditions. We first transform the original problem into a standard quadratic programming structure with affine equality constraints:

$$\min_w \|Aw - b\|_2^2 + \lambda w^\top U w, \text{ s.t. } \mathbf{1}^\top w = 1, w \geq 0, \tag{11}$$

where $A = [\Gamma_1, \ldots, \Gamma_K] \in \mathbb{R}^{2C \times K}$ and $b = [\Gamma_t] \in \mathbb{R}^{2C}$ denote the vectorized feature statistics of the asset library and the target domain. $U = \text{diag}\{u_1, \cdots, u_k\}$ denotes the diagonal matrix of uncertainty scores.

We further solve this linearly constrained quadratic program via the KKT conditions. Specifically, we incorporate the affine constraint into the objective via a Lagrange multiplier $\nu$. The stationarity condition implies that the gradient of the objective must be aligned with the constraint normal. Let $\mathcal{H} = A^\top A + \lambda U$ denote the regularized Hessian matrix capturing asset correlations, and $g = A^\top b$ denotes the projection of the target onto the asset basis. Solving the resulting linear system yields the optimal weights:

$$w^* = \mathcal{H}^{-1}(g - \nu \mathbf{1}), \quad \nu = \frac{\mathbf{1}^\top \mathcal{H}^{-1} g - 1}{\mathbf{1}^\top \mathcal{H}^{-1} \mathbf{1}}. \tag{12}$$

This derivation allows IDEA to construct an optimal bridge via simple matrix operations. By eliminating the need for iterative tuning, this mechanism acts as a training-free shortcut to the target domain, significantly accelerating the adaptation process by bypassing the lengthy trajectory.

### 4.3. Overall Procedure of IDEA

At each navigation step, IDEA dynamically determines whether to utilize the bridge or acquire a new asset based on domain coverage. Specifically, we first solve for the optimal weights $w$ using Eq. 12 and construct the adaptation bridge $P_b(w)$. Next, we measure the statistics discrepancy with and without prompt injection, denoted by $d_p$ and $d_0$, respectively. If the reduction satisfies $d_p < \tau \cdot d_0$, the domain is deemed covered and $P_b(w)$ is applied for inference.

Otherwise, we regard it as a novel domain and optimize $P_b(w)$ via Eq. 5, yielding a new asset $\mathcal{A}^* = \{P^*, \Gamma^*, u^*\}$. To maintain a fixed budget, we employ a nearest-neighbor merging strategy:

$$\begin{cases} \mathcal{A}_k \leftarrow \frac{1}{2}\left(\mathcal{A}_k + \mathcal{A}^*\right), & \text{if } K \geq K_{\max}; \\ \mathcal{M} \leftarrow \mathcal{M} \cup \{\mathcal{A}^*\}, & \text{if } K < K_{\max}; \\ \text{where } k = \arg\min_k d(\Gamma_k, \Gamma^*), \end{cases} \tag{13}$$

where $K_{max}$ is a capacity budget of the asset library.

### 4.4. Theoretical Analysis on Stability and Generalization

We provide theoretical insight into why our method can stably reduce generalization risk on novel target domains. Under the common assumption that feature representations within a domain follow multivariate Gaussian distributions (Heusel et al., 2017; Hu et al., 2025a), we establish two key results: 1) The mixture weights produced by Eq. 10 tighten a principled upper bound on the target generalization error, and are optimal within the convex hull of historical assets. 2) The solution in Eq. 12 is Lipschitz-stable with respect to perturbations in the estimated statistics, implying robustness to test-time estimation noise and bias. Together, these propositions demonstrate that our bridge can stably reduce generalization risk at test time without additional training—a crucial advantage for VLN deployments.

All detailed analysis and proof are provided in Appendix. A.

## 5. Experiments

### 5.1. Experimental Setup

For a fair comparison, we follow the evaluation pipeline in prior works (Gao et al., 2024; Kim et al., 2025), including datasets, pre-trained policies, training recipes, and evaluation protocols.

**Datasets.** We evaluate our method on three representative VLN benchmarks covering both discrete and continuous environments: REVERIE (Qi et al., 2020), R2R (Anderson et al., 2018), and R2R-CE (Krantz et al., 2020). REVERIE is a goal-oriented task where agents localize remote objects based on high-level instructions. R2R is a standard instruction-following benchmark, where agents navigate to a target viewpoint via step-by-step instructions. Beyond these discrete settings, we also adopt R2R-CE, a continuous variant of R2R that introduces low-level control challenges.

**Pre-trained Navigation Policies.** We validate our method on four base models: HAMT (Chen et al., 2021), DUET (Chen et al., 2022c), BEVBert (An et al., 2023), and ETPNav (An et al., 2024). HAMT employs a transformer architecture optimized via reinforcement learning to handle long-horizon navigation. DUET advances this by integrating a topological map into a dual-scale graph

*Table 1.* Experimental results for different TTA strategies on REVERIE datasets. We highlight the best and second results with **bold** and underline respectively. In the last column, we report the average per-episode inference time, measured in milliseconds.

| Methods+Model | REVERIE Val seen | | | | REVERIE Val unseen | | | | REVERIE Test unseen | | | | Time(ms) |
|---|---|---|---|---|---|---|---|---|---|---|---|---|---|
| | OSR↑ | SR↑ | SPL↑ | RGSPL↑ | OSR↑ | SR↑ | SPL↑ | RGSPL↑ | OSR↑ | SR↑ | SPL↑ | RGSPL↑ | |
| HAMT (Chen et al., 2021) | 47.65 | 43.29 | 40.19 | 25.18 | 36.84 | 32.95 | 30.20 | 17.28 | 33.41 | 30.40 | 26.67 | 13.08 | 74.9 |
| + Tent (Wang et al., 2021) | 46.03 | 43.43 | 40.78 | 25.81 | 32.60 | 30.56 | 28.23 | 14.48 | 25.06 | 23.73 | 21.78 | 10.82 | 147.7 |
| + SAR (Niu et al., 2023) | 47.12 | 43.85 | 39.97 | 25.44 | 32.86 | 31.12 | 29.15 | 15.23 | 27.94 | 25.86 | 23.08 | 11.58 | 197.25 |
| + ViDA (Liu et al., 2024b) | 47.67 | 43.55 | 41.23 | 25.27 | 32.74 | 30.97 | 28.82 | 14.97 | 27.03 | 24.81 | 22.45 | 11.23 | $5.49 \times 10^3$ |
| + FSTTA (Gao et al., 2024) | 48.21 | 42.87 | 39.56 | 24.58 | 36.78 | 32.89 | 30.51 | 17.20 | 33.39 | 30.39 | 26.65 | 13.61 | 613.4 |
| + ReCAP (Hu et al., 2025b) | 48.49 | 44.06 | 40.69 | 25.46 | 37.04 | 33.06 | 30.28 | 17.37 | 34.11 | 30.51 | 24.27 | 13.11 | 213.5 |
| **+ IDEA (Ours)** | **50.67** | **47.33** | **42.13** | **26.82** | **39.87** | **34.92** | **31.52** | **17.76** | **38.14** | **32.81** | **28.52** | **14.45** | 245.8 |
| DUET (Chen et al., 2022c) | 73.86 | 71.75 | 63.94 | 51.14 | 51.07 | 46.98 | 33.73 | 23.03 | 56.91 | 52.51 | 36.06 | 22.06 | 123.2 |
| + Tent (Wang et al., 2021) | 73.72 | 71.89 | 64.06 | 50.41 | 51.43 | 47.55 | 33.99 | 23.32 | 57.12 | 52.61 | 36.17 | 22.16 | 258.9 |
| + SAR (Niu et al., 2023) | 74.84 | 71.75 | 64.43 | 51.70 | 53.26 | 48.00 | 33.92 | 23.09 | 57.11 | 53.04 | 36.07 | 22.27 | 267.5 |
| + ViDA (Liu et al., 2024b) | 73.99 | 72.49 | 63.94 | 50.89 | 52.53 | 48.14 | 32.45 | 21.92 | 56.78 | 52.74 | 35.10 | 21.77 | $7.31 \times 10^3$ |
| + FSTTA (Gao et al., 2024) | 75.59 | 75.48 | 65.84 | 52.23 | 56.26 | 54.15 | 36.41 | 23.56 | 58.44 | 53.40 | 36.43 | 22.40 | 833.6 |
| + ReCAP (Hu et al., 2025b) | 75.06 | 74.72 | 65.87 | 53.42 | 56.67 | 54.74 | 36.22 | 23.74 | 57.72 | 53.07 | 36.52 | 22.47 | 359.4 |
| **+ IDEA (Ours)** | **78.45** | **78.24** | **67.74** | **55.07** | **58.51** | **56.92** | **38.03** | **25.47** | **58.91** | **55.12** | **39.84** | **24.52** | 344.3 |

*Table 2.* Experimental results on the R2R dataset.

| Methods | R2R Val Seen | | | | R2R Val Unseen | | | |
|---|---|---|---|---|---|---|---|---|
| | TL↓ | NE↓ | SR↑ | SPL↑ | TL↓ | NE↓ | SR↑ | SPL↑ |
| DUET | 12.33 | 2.28 | 79 | 73 | 13.94 | 3.31 | 72 | 60 |
| + Tent | 12.17 | 2.38 | 78 | 72 | 13.78 | 3.42 | 72 | 60 |
| + SAR | 12.05 | 2.28 | 78 | 72 | 13.59 | 3.28 | 72 | 61 |
| + ViDA | 12.11 | 2.31 | 79 | 73 | 13.63 | 3.34 | 72 | 61 |
| + FSTTA | 13.39 | 2.25 | 79 | 73 | 14.64 | 3.03 | 75 | 62 |
| + ReCAP | 12.02 | 2.27 | 78 | 73 | 13.39 | 3.28 | 72 | 61 |
| **+ IDEA** | **11.23** | **2.03** | **81** | **76** | **12.47** | **2.91** | **76** | **67** |
| BEVBert | 13.56 | 2.17 | 81 | 74 | 14.55 | 2.81 | 75 | 64 |
| + Tent | 12.68 | 2.36 | 80 | 74 | 13.14 | 2.93 | 74 | 63 |
| + SAR | 12.49 | 2.28 | 80 | 74 | 12.98 | 2.79 | 75 | 64 |
| + ViDA | 12.53 | 2.31 | 81 | 74 | 13.11 | 2.83 | 75 | 64 |
| + FSTTA | 12.28 | 2.31 | 80 | 75 | 13.96 | 2.83 | 74 | 63 |
| + ReCAP | 12.31 | 2.27 | 81 | 74 | 12.94 | 2.78 | 75 | 64 |
| **+ IDEA** | **10.89** | **2.15** | **83** | **79** | **12.03** | **2.53** | **76** | **68** |

*Table 3.* Experimental results on the R2R-CE dataset.

| Methods | R2R-CE Val Seen | | | | | R2R-CE Val Unseen | | | | |
|---|---|---|---|---|---|---|---|---|---|---|
| | TL↓ | NE↓ | OSR↑ | SR↑ | SPL↑ | TL↓ | NE↓ | OSR↑ | SR↑ | SPL↑ |
| BEVBert | 13.98 | 3.77 | 73 | 68 | 60 | 13.27 | 4.57 | 67 | 59 | 50 |
| + Tent | 12.74 | 3.35 | 76 | 70 | 62 | 13.29 | 4.61 | 65 | 59 | 49 |
| + SAR | 12.53 | 3.28 | 76 | 70 | 63 | 13.23 | 4.54 | 66 | 59 | 50 |
| + ViDA | 12.61 | 3.31 | 76 | 71 | 62 | 13.26 | 4.57 | 67 | 59 | 49 |
| + FSTTA | 14.07 | 4.11 | 74 | 69 | 60 | 13.11 | 4.39 | 65 | 60 | 51 |
| + ReCAP | 12.40 | 3.31 | 76 | 71 | 63 | 13.01 | 4.57 | 66 | 60 | 50 |
| **+ IDEA** | **12.27** | **3.04** | **78** | **73** | **64** | **12.67** | **4.26** | **69** | **62** | **52** |
| ETPNav | 11.78 | 3.95 | 72 | 66 | 59 | 11.99 | 4.71 | 65 | 57 | 49 |
| + Tent | 11.36 | 3.94 | 72 | 66 | 59 | 11.57 | 4.74 | 64 | 57 | 49 |
| + SAR | 11.31 | 3.89 | 72 | 66 | 60 | 11.55 | 4.67 | 65 | 57 | 49 |
| + ViDA | 11.58 | 3.91 | 72 | 66 | 60 | 11.62 | 4.72 | 64 | 57 | 49 |
| + FSTTA | 11.35 | 3.93 | 72 | 66 | 59 | 11.57 | 4.77 | 64 | 57 | 49 |
| + ReCAP | 11.31 | 3.92 | 72 | 66 | 60 | 11.56 | 4.74 | 65 | 57 | 50 |
| **+ IDEA** | **10.87** | **3.82** | **73** | **68** | **62** | **11.17** | **4.47** | **67** | **59** | **51** |

transformer for efficient global planning. BEVBert further enhances spatial reasoning by incorporating bird's-eye-view representations. For continuous environments, we utilize ETPNav, which is designed for robust long-range planning under continuous control. Our IDEA framework is applied to these pre-trained policies during inference phase.

**Compared Methods.** We compare our IDEA with the following state-of-the-art methods: SAR (Niu et al., 2023) employs entropy minimization combined with sharpness-aware minimization to enhance stability. ViDA (Liu et al., 2024b) injects low-rank and high-rank adapters to decouple domain-invariant and domain-specific representations. FSTTA (Gao et al., 2024) performs decomposition-accumulation analysis for both gradients and parameters. ReCAP (Hu et al., 2025b) models regional uncertainty, enforcing implicit data scaling. Additionally, we provide a detailed comparison with feedback-driven methods in the Appendix. B.

**Implementation Details.** We adopt a batch size of 1 to properly simulate the online streaming setting and employ the AdamW optimizer with a fixed learning rate of $3 \times 10^{-3}$ and a momentum of $0.9$. Regarding hyperparameters, we set the prompt length $L = 4$, the asset library size $K_{max} = 32$, the regularization strength $\lambda = 0.4$, and ratio threshold $\tau = 0.7$ by default. All experiments are conducted on a single NVIDIA RTX 4090 GPU.

**Evaluation Protocols.** We follow the commonly used evaluation protocols from the previous works (Chen et al., 2022c;b; Li et al., 2022; Wang et al., 2023): Success Rate (SR), Oracle Success Rate (OSR), Success weighted by Path Length (SPL), Remote Grounding Success weighted by Path Length (RGSPL), Trajectory Length (TL), and Navigation Error (NE). Detailed definitions are provided in Appendix. E.

### 5.2. Main Results

**Evaluation on REVERIE.** We first compare our IDEA with previous methods on the REVERIE dataset, where TTA is applied to HAMT and DUET. The results, reported in Tab. 1, reveal several key observations: 1) Existing methods struggle on the test unseen split, where nearly all competitors fail to improve or even degrade the SPL metric, indicating severe instability and negative transfer. 2) In contrast, IDEA consistently outperforms compared approaches across all data splits, being the *only* method to achieve positive gains on all metrics. Notably, it yields substantial average gains of **+2.5%** SR and **+2.8%** SPL on test unseen. 3) Regarding efficiency, IDEA requires less than half the inference time of FSTTA and is comparable to ReCAP, while delivering more stable gains. As shown in Fig. 4, this efficiency stems from training-free adaptation bridge, which bypasses iterative optimization for the majority of incoming domains, validating our design intuition of asset reuse.

*Table 4.* Experimental results for asset transfer in our IDEA framework on REVERIE dataset. We highlight the best result with **bold**.

| | Shared Assets | Adaptation | Val seen → Test unseen | | | | Test unseen → Val unseen | | | | Val seen + unseen → Test unseen | | | |
|---|---|---|---|---|---|---|---|---|---|---|---|---|---|---|
| | | | OSR↑ | SR↑ | SPL↑ | RGSPL↑ | OSR↑ | SR↑ | SPL↑ | RGSPL↑ | OSR↑ | SR↑ | SPL↑ | RGSPL↑ |
| HAMT | ✘ | ✘ | 33.41 | 30.40 | 26.67 | 13.08 | 36.84 | 32.95 | 30.20 | 17.28 | 33.41 | 30.40 | 26.67 | 13.08 |
| | ✘ | ✔ | 38.14 | 32.81 | 28.52 | 14.45 | 39.87 | 34.92 | 31.52 | 17.76 | 38.14 | 32.81 | 28.52 | 14.45 |
| | ✔ | ✘ | 33.83 | 30.88 | 27.05 | 13.38 | 37.35 | 33.57 | 30.77 | 17.39 | 34.57 | 31.68 | 27.74 | 13.81 |
| | ✔ | ✔ | **38.31** | **33.07** | **28.63** | **14.69** | **40.18** | **35.27** | **31.94** | **18.11** | **38.51** | **33.32** | **28.83** | **14.84** |
| DUET | ✘ | ✘ | 56.91 | 52.51 | 36.06 | 22.06 | 51.07 | 46.98 | 33.73 | 23.03 | 56.91 | 52.51 | 36.06 | 22.06 |
| | ✘ | ✔ | 58.91 | 55.12 | 39.84 | 24.52 | 58.51 | 56.92 | 38.03 | 25.47 | 58.91 | 55.12 | 39.84 | 24.52 |
| | ✔ | ✘ | 57.38 | 53.12 | 36.57 | 23.61 | 51.38 | 47.45 | 33.96 | 23.27 | 57.83 | 53.44 | 36.91 | 24.58 |
| | ✔ | ✔ | **59.23** | **55.87** | **40.08** | **24.85** | **58.83** | **57.39** | **38.49** | **25.84** | **59.44** | **56.11** | **40.35** | **25.47** |

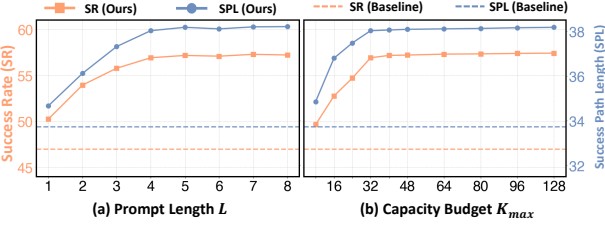

*Figure 3.* (a) Performance with varying length $L$ of the soft prompt. (b) Performance with different capacity $K_{max}$ of the asset library.

**Evaluation on R2R & R2R-CE.** We further evaluate different methods on R2R and R2R-CE datasets, as shown in Tab. 2&3. The experimental results yield the following observations: 1) On the R2R val seen and R2R-CE val unseen splits, existing methods achieve only marginal improvements, whereas our method boosts performance by **+2% SR** and **+3% SPL**. 2) IDEA consistently enhances two different base models and maintains the best performance across all four scenarios, further demonstrating its strong generalization across discrete and continuous environments.

### 5.3. Experiments on Asset Transfer

To evaluate asset transferability, we simulate a deployment scenario where libraries constructed in source environments are transferred to target agents. We design three protocols on the REVERIE dataset: *Val Seen → Test Unseen*, *Test Unseen → Val Unseen*, and *Val Seen + Unseen → Test Unseen*. The third protocol represents a multi-source setting, where assets collected by multiple agents are aggregated for the new user.

We compare four configurations based on the availability of shared assets and the usage of online adaptation, as shown in Tab. 4. The results yield three significant insights: 1) **Training-free Gain:** Shared assets provide consistent training-free improvements across all scenarios. 2) **Synergy with Adaptation:** Pre-loaded assets complement online adaptation without conflict, achieving peak performance when combined. 3) **Scaling Benefit**: The multi-source setting confirms that aggregating larger libraries facilitates robust bridge construction, boosting SR by +1.2% and SPL by +1.0%. This validates IDEA's potential for a collaborative knowledge-sharing paradigm, where aggregating assets from diverse users continuously enhances generalization.

Additionally, we investigate cross-task transferability

(REVERIE ↔ R2R) to validate the versatility of assets. Due to space constraints, results are detailed in the Appendix. B.

## 6. Ablation Study and Visualization

In this section, without loss of generality, we conduct ablation studies and visualizations using DUET on the REVERIE valid unseen split as a representative testbed. Focusing on two pivotal components of IDEA, *Domain Asset Library* and *Bridge Construction*, we perform various experiments to analyze their impacts. Please refer to the Appendix. C for more experiments and detailed analysis.

### 6.1. Sensitivity Analysis of Asset Configurations

We assess the robustness of IDEA with respect to two critical asset configurations: the prompt length $L$ and the library capacity budget $K_{max}$. We conduct ablation experiments on these two key coefficients independently:

As shown in Fig. 3(a), increasing the prompt length yields substantial performance gains on both SR and SPL, as longer prompts offer greater capacity to encode transferable knowledge. However, when $L$ exceeds the optimal range (*e.g.*, 4), further extension yields only marginal improvements. Therefore, we set $L$ to 4 by default to balance representation power and efficiency. Similarly, in Fig. 3(b), navigation performance improves rapidly as the library capacity expands, reaching a robust plateau at $K_{max} = 32$. This confirms that a denser set of historical assets facilitates more precise bridge construction. Extending the capacity beyond this threshold results in diminishing returns, indicating that the library has sufficiently covered the representative modes of test environments. Therefore, we set $K_{max}$ to 32 by default. Notably, under the default setting, IDEA incurs a negligible memory overhead of only **0.58 MB**, adding less than **0.1%** to the parameter count of the pre-trained model.

### 6.2. Effectiveness of Components.

We investigate the impact of individual components by comparing the full method with partial variations, as shown in Tab. 5. Here, *Dec.* represents the widely-adopted baseline using layer-wise decay weighting (Bao et al., 2022), while *Ret.* denotes a hard selection strategy that retrieves the single nearest neighbor asset as the bridge. Our analysis

*Table 5.* Impact of component variations. **Fsh./Brg.** denote our Fisher-guided weighting (Eq. 8) and closed-form bridge solution (Eq. 12). For comparison, **Dec./Ret.** denote the widely-adopted layer-wise decay weighting (Bao et al., 2022) and the retrieval of the nearest asset. **Src.** denotes source model without adaptation.

| Src. | Fsh. | Dec. | Brg. | Ret. | REVERIE Val unseen | | | | REVERIE Test unseen | | | |
|---|---|---|---|---|---|---|---|---|---|---|---|---|
| | | | | | OSR | SR | SPL | RGSPL | OSR | SR | SPL | RGSPL |
| ✓ | | | | | 51.07 | 46.98 | 33.73 | 23.03 | 56.91 | 52.51 | 36.06 | 22.06 |
| | ✓ | | | ✓ | 54.88 | 50.47 | 35.53 | 24.18 | 57.89 | 54.11 | 37.92 | 22.82 |
| | | ✓ | | ✓ | 50.87 | 47.65 | 33.58 | 22.94 | 56.98 | 52.72 | 36.23 | 22.14 |
| | | ✓ | ✓ | | 51.47 | 47.88 | 34.30 | 23.49 | 57.31 | 52.83 | 36.87 | 22.42 |
| | ✓ | | ✓ | | **58.51** | **56.92** | **38.03** | **25.47** | **58.91** | **55.12** | **39.84** | **24.52** |

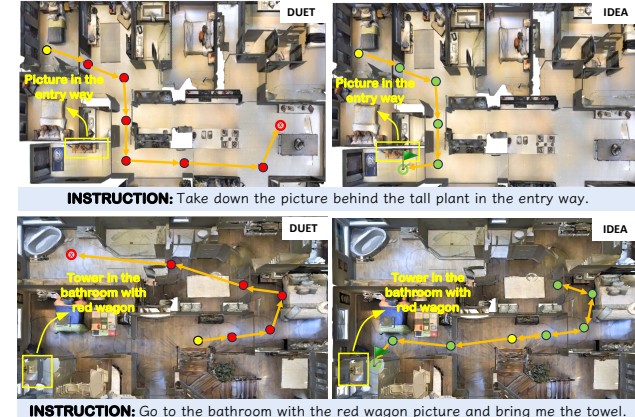

*Figure 4.* Visualization of adaptation process. (a) Asset accumulation & Coverage. (b) Discrepancy reduction by our bridge.

yields three key observations: 1) The naive baseline employing nearest asset retrieval with standard decay weighting yields performance that merely fluctuates around the unadapted source model. This indicates that hard selection based on heuristic weighting fails to address domain shifts, often resulting in suboptimal prior retrieval. 2) Fisher-guided weighting brings obvious improvements of $+3.5\%$ SR, validating that distinguishing functional layer parameters is critical for identifying transferable knowledge. 3) Our closed-form bridge demonstrates clear superiority over hard retrieval, underscoring the importance of optimal projection for constructing a robust adaptation shortcut.

### 6.3. Complementary Effects

As shown in Tab. 5, our complete framework consistently achieves the best navigation performance, demonstrating the compatibility of different components. To further investigate the complementary effects of our asset-bridge design, we visualize the adaptation process. As shown in Fig. 4(a), we observe a positive correlation between library growth and domain coverage (*i.e.* incoming domain where the bridge successfully reduces the discrepancy below the threshold). This validates that a richer repository of "building blocks" facilitates bridge construction across a wider range of distribution shifts. As shown in Fig. 4(b), the injected bridge significantly reduces the statistical divergence from the target domain. By narrowing the domain gap, the bridge alleviates the burden of subsequent asset optimization, providing a superior initialization for learning novel environments.

### 6.4. Qualitative Results

Based on the qualitative comparisons in Fig. 5, IDEA significantly enhances the agent's ability to identify discriminative landmarks compared to the baseline DUET. In challenging scenarios, such as finding a specific room with a *"red*

*Figure 5.* Qualitative comparison of navigation results on REVERIE. Yellow points denote starting positions. Directed lines trace the predicted paths, ending in green (success) or red (failure).

*wagon"* (bottom), the baseline fails to ground fine-grained textual cues within the novel environment, leading to incorrect termination. In contrast, IDEA successfully corrects these deviations by capturing task-essential visual features. These visualizations further confirm that our adaptation bridge effectively aligns the agent's perception with complex unseen environments, achieving robust navigation that precisely matches the instruction.

## 7. Conclusion

In this paper, we propose IDEA, a novel framework designed to transform adaptation knowledge into reusable assets. We develop a Fisher-guided weighting scheme to capture transferable knowledge for effective asset utilization, and further construct a training-free adaptation bridge to bypass the lengthy optimization process. We demonstrate the consistent effectiveness of IDEA across diverse navigation benchmarks and model architectures. We hope this work inspires the future research on lifelong experience accumulation, paving the way for constructing standardized and generalized asset protocols. Such mechanisms are pivotal for enabling efficient deployment and fostering collaboration among agents operating in dynamic environments.

**Limitations.** First, our evaluations focus on single-agent benchmarks. While this setting effectively validates individual adaptability, the potential of IDEA framework in multi-agent scenarios remains underexplored. In future work, we plan to extend IDEA to collaborative frameworks, investigating mechanisms for efficient asset sharing and merging among heterogeneous agents. Second, the constructed assets are currently coupled with specific feature spaces. While IDEA demonstrates robust cross-domain and cross-dataset transferability, this dependency restricts its direct application across different model architectures. In future work, we will explore architecture-agnostic asset protocols to break this barrier and establish a more universal standard.

## Acknowledgements

This work was supported by the PKU-NTU Joint Research Institute (JRI) sponsored by a donation from the Ng Teng Fong Charitable Foundation, in part by AI Joint Lab of Future Urban Infrastructure sponsored by Fuzhou Chengtou New Infrastructure Group and Boyun Vision Co. Ltd.

## Impact Statement

This paper presents work whose goal is to advance the field of Test-Time Adaptation. Its societal impact lies primarily in enabling Vision-Language Navigation agents to operate reliably in real-world dynamic scenarios without the need for resource-intensive retraining. By introducing the training-free adaptation bridge based on assets, our work significantly reduces the computational latency associated with online adaptation. This facilitates the deployment of intelligent agents on resource-constrained edge devices, such as domestic service robots. Ethically, it promotes sustainable AI practices by eliminating the energy consumption of iterative gradient updates, aligning with broader goals of developing efficient, eco-friendly autonomous systems.

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

# Turning Adaptation into Assets: Cross-Domain Bridging for Online Vision-Language Navigation

## ————Appendix————

The structure of Appendix is as follows:

- Appendix A contains all missing proofs and propositions for theoretical claims in the main manuscript.

- Appendix E details the datasets and evaluation metrics used in our experiments.

## A. Theoretical Proof

Below, we will provide detailed proofs of the theoretical results in the main manuscript.

**Notation.** First, we recall the notation used in the main paper and this appendix: $\pi_\theta$ denotes the policy model parameterized by $\theta$, with $\pi_\theta(a \mid z)$ representing the output distribution over actions. $\mathcal{Z}_t = \{z_i\}_{i=1}^N \in \mathbb{R}^{N \times C}$ denotes the set of multi-modal representations, where $z_i$ is the fused feature for the $i$-th candidate node generated by the fusion function $\phi(\cdot)$. Here, $N$ is the number of candidate nodes and $C$ is the feature dimension. $P := \{p_i\}_{i=1}^L$ denotes the set of injected prompts. Regarding domain statistics, $\Gamma_S^{(\ell)} = \{\mu_S^{(\ell)}, \sigma_S^{(\ell)}\}$ denotes the source statistics at the $\ell$-th layer of the fusion transformer (where $M$ is the total number of layers). Similarly, $\Gamma_t(P)$ and $\Gamma_t$ denote the target statistics computed with and without the prompt $P$, respectively. Finally, $P_b(w) = \sum_{j=1}^K w_j P_j$ denotes the composite bridge prompt weighted by $w$, and $\Gamma_b(w) = \sum_{j=1}^K w_j \Gamma_j$ denotes the corresponding bridge statistics, where the subscript $b$ indicates the bridge domain.

### A.1. Fisher Proxy Derivation

In this section, we establish the connection between the curvature of the policy objective and the Fisher Information Matrix in Sec. 4.1. We show that under the expectation over the policy distribution, the computationally expensive Hessian can be approximated by the first-order Fisher Information Matrix.

**Proposition A.1.** *Let $\pi_\theta(a \mid z)$ be a differentiable policy distribution on representation $z$, and let the loss function $\mathcal{L}(z) = -\log \pi_\theta(a \mid z)$ be defined as the negative log-likelihood. Under standard regularity conditions permitting the exchange of differentiation and integration, the expected Hessian of the loss with respect to $z$ is equivalent to the Fisher Information Matrix $\Phi(z)$. Specifically:*

$$\mathbb{E}_{a \sim \pi_\theta(\cdot \mid z)} \left[ \nabla_z^2 \left( -\log \pi_\theta(a \mid z) \right) \right] = \mathbb{E}_{a \sim \pi_\theta(\cdot \mid z)} \left[ \nabla_z \log \pi_\theta(a \mid z) \nabla_z \log \pi_\theta(a \mid z)^\top \right]. \tag{14}$$

*Proof.* Let the Hessian of the negative log-likelihood for a specific action $a$ be denoted as $H(z; a) = \nabla_z^2 \left( -\log \pi_\theta(a \mid z) \right)$. We analyze the curvature by explicitly expanding the second-order derivative. First, using the chain rule, the gradient of the log-likelihood (the score function) is given by:

$$\nabla_z \log \pi_\theta(a \mid z) = \frac{\nabla_z \pi_\theta(a \mid z)}{\pi_\theta(a \mid z)}. \tag{15}$$

By differentiating Eq. (15) with respect to $z$ again, we apply the quotient rule for vector-valued functions. Note that for the

negative log-likelihood, the sign is inverted:

$$
\begin{aligned}
H(z; a) &= \nabla_z \left( -\frac{\nabla_z \pi_\theta(a \mid z)}{\pi_\theta(a \mid z)} \right) \\
&= -\frac{\pi_\theta(a \mid z)\nabla_z^2 \pi_\theta(a \mid z) - \nabla_z \pi_\theta(a \mid z)\nabla_z \pi_\theta(a \mid z)^\top}{\pi_\theta(a \mid z)^2} \\
&= \underbrace{\frac{\nabla_z \pi_\theta(a \mid z)}{\pi_\theta(a \mid z)} \left( \frac{\nabla_z \pi_\theta(a \mid z)}{\pi_\theta(a \mid z)} \right)^\top}_{\text{Term A}} - \underbrace{\frac{\nabla_z^2 \pi_\theta(a \mid z)}{\pi_\theta(a \mid z)}}_{\text{Term B}}.
\end{aligned}
\tag{16}
$$

Substituting the score function definition from Eq. (15) back into Term A, we observe that the point-wise Hessian decomposes into the outer product of gradients minus a normalized curvature term:

$$
H(z; a) = \nabla_z \log \pi_\theta(a \mid z)\nabla_z \log \pi_\theta(a \mid z)^\top - \frac{\nabla_z^2 \pi_\theta(a \mid z)}{\pi_\theta(a \mid z)}.
\tag{17}
$$

Next, we take the expectation of $H(z; a)$ with respect to the policy distribution $a \sim \pi_\theta(\cdot \mid z)$. Due to the linearity of expectation, we analyze the two terms separately. For Term B, we invoke the standard regularity conditions that permit the interchange of differentiation and integration (Leibniz integral rule). Consequently, the expectation of the second term vanishes:

$$
\begin{aligned}
\mathbb{E}_{a \sim \pi_\theta(\cdot \mid z)} \left[ \frac{\nabla_z^2 \pi_\theta(a \mid z)}{\pi_\theta(a \mid z)} \right] &= \int \pi_\theta(a \mid z) \left( \frac{\nabla_z^2 \pi_\theta(a \mid z)}{\pi_\theta(a \mid z)} \right) da \\
&= \int \nabla_z^2 \pi_\theta(a \mid z) da \\
&= \nabla_z^2 \left( \int \pi_\theta(a \mid z) da \right) \\
&= \nabla_z^2 (1) = 0.
\end{aligned}
\tag{18}
$$

Since the probability density function integrates to 1, its gradient (and Hessian) with respect to parameters $z$ must be zero. Therefore, the expected Hessian is determined solely by the first term:

$$
\mathbb{E}_{a \sim \pi_\theta(\cdot \mid z)}[H(z; a)] = \mathbb{E}_{a \sim \pi_\theta(\cdot \mid z)} \left[ \nabla_z \log \pi_\theta(a \mid z)\nabla_z \log \pi_\theta(a \mid z)^\top \right] = \Phi(z).
\tag{19}
$$

This concludes the proof. $\qquad\square$

## A.2. Closed-form Solution of Adaptation Bridge

In this subsection, we analyze the optimization landscape of the proposed objective. By approximating the Wasserstein distance with the Euclidean distance of feature statistics and incorporating the uncertainty-aware regularization, we derive the optimal mixture weights analytically.

**Proposition A.2.** *(Optimal Mixture Weights) Consider the objective function defined by the regularized matching of feature statistics:*

$$
\min_{w \in \mathbb{R}^K} \mathcal{J}(w) = \|Aw - b\|_2^2 + \lambda w^\top U w, \quad subject\ to \quad \mathbf{1}^\top w = 1,
$$

*where $A \in \mathbb{R}^{2C \times K}$ aggregates the asset statistics, $b \in \mathbb{R}^{2C}$ represents the target statistics, and $U$ is a diagonal matrix of uncertainty scores. Let $\mathcal{H} = A^\top A + \lambda U$ be the regularized Hessian and $g = A^\top b$ be the projection vector. The optimal solution $w^*$ is given in closed form by:*

$$
w^* = \mathcal{H}^{-1}(g - \nu \mathbf{1}), \quad with \quad \nu = \frac{\mathbf{1}^\top \mathcal{H}^{-1} g - 1}{\mathbf{1}^\top \mathcal{H}^{-1} \mathbf{1}}.
$$

*Proof.* The optimization problem represents a convex quadratic program (QP) with a linear equality constraint. We proceed by the method of Lagrange multipliers. The objective function $\mathcal{J}(w)$ can be expanded as:

$$
\begin{aligned}
\mathcal{J}(w) &= (Aw - b)^\top (Aw - b) + \lambda w^\top U w \\
&= w^\top A^\top A w - 2b^\top A w + b^\top b + \lambda w^\top U w \\
&= w^\top (A^\top A + \lambda U) w - 2(A^\top b)^\top w + b^\top b.
\end{aligned}
\tag{20}
$$

Substituting the definitions of $\mathcal{H} = A^\top A + \lambda U$ and $g = A^\top b$, and discarding the constant term $b^\top b$ which does not affect the optimization. To facilitate gradient derivation, we consider the equivalent optimization problem scaled by a factor of $1/2$:

$$
\min_w \quad \frac{1}{2} w^\top \mathcal{H} w - g^\top w \quad \text{s.t.} \quad \mathbf{1}^\top w = 1.
\tag{21}
$$

This is a convex quadratic programming problem with a single affine equality constraint. We construct the Lagrangian function $\mathcal{L}(w, \nu)$ with a scalar Lagrange multiplier $\nu$ associated with the equality constraint:

$$
\mathcal{L}(w, \nu) = \frac{1}{2} w^\top \mathcal{H} w - g^\top w + \nu(\mathbf{1}^\top w - 1).
\tag{22}
$$

According to the Karush–Kuhn–Tucker (KKT) conditions for this convex problem, the optimal point $(w^*, \nu^*)$ must satisfy the stationarity condition $\nabla_w \mathcal{L} = 0$ and the primal feasibility condition $\nabla_\nu \mathcal{L} = 0$.

**Stationarity Condition.** Taking the gradient with respect to $w$ and setting it to zero:

$$
\nabla_w \mathcal{L}(w, \nu) = \mathcal{H} w - g + \nu \mathbf{1} = 0.
\tag{23}
$$

Rearranging terms to solve for $w$:

$$
\mathcal{H} w = g - \nu \mathbf{1} \implies w = \mathcal{H}^{-1}(g - \nu \mathbf{1}).
\tag{24}
$$

Here, $H$ is invertible (positive definite) because $A^\top A$ is positive semi-definite and $\lambda U$ is positive definite.

**Primal Feasibility.** We impose the constraint $\mathbf{1}^\top w = 1$ on the solution derived in Eq. (24):

$$
\mathbf{1}^\top \mathcal{H}^{-1}(g - \nu \mathbf{1}) = 1.
\tag{25}
$$

Distributing the vector multiplication:

$$
\mathbf{1}^\top \mathcal{H}^{-1} g - \nu(\mathbf{1}^\top \mathcal{H}^{-1} \mathbf{1}) = 1.
\tag{26}
$$

**Solving for the Dual Variable.** Solving the algebraic equation above for the scalar $\nu$:

$$
\nu(\mathbf{1}^\top \mathcal{H}^{-1} \mathbf{1}) = \mathbf{1}^\top \mathcal{H}^{-1} g - 1 \implies \nu = \frac{\mathbf{1}^\top \mathcal{H}^{-1} g - 1}{\mathbf{1}^\top \mathcal{H}^{-1} \mathbf{1}}.
\tag{27}
$$

Finally, substituting this value of $\nu$ back into Eq. (24) yields the closed-form solution $w^*$, thereby concluding the proof. $\square$

**Remark (Non-Negativity and Implementation).** Although Proposition A.2 derives the solution for the equality-constrained relaxation, the non-negativity constraint ($w \geq 0$) is theoretically essential. It ensures that the bridge distribution remains a valid *convex combination* of the source assets, constraining the adaptation to the convex hull of the library $\mathcal{M}$. Allowing negative weights would imply improper extrapolation, potentially leading to unstable adaptation behavior. In practice, the solution exhibits a strong tendency toward positivity due to the regularization structure. In rare cases where the solution $w^*$ violates the non-negativity constraint, we apply a tractable projection step: $\hat{w} = w^* / \|w^*\|_1$ after clipping negative values to zero ($\hat{w} \leftarrow \max(0, w^*)$). This strategy preserves the $\mathcal{O}(K^3)$ computational efficiency of the closed-form solution while strictly satisfying the simplex constraints.

### A.3. Generalization Bound Analysis

In this subsection, we provide a theoretical analysis of the proposed uncertainty-aware adaptation bridge. We show that the optimization objective in Eq. 10 serves as a surrogate for minimizing a certified upper bound of the expected target risk. *Remark: For clarity and alignment with standard learning theory literature, we adopt standard notations in this subsection, which may differ from those in previous sections.*

**Setup and Notations.** Let $\mathcal{D}_t$ be the target distribution over $(x, y) \in \mathcal{X} \times \mathcal{Y}$. Let $\phi : \mathcal{X} \to \mathbb{R}^d$ be a fixed representation map (*e.g.*, a frozen backbone) and denote $z = \phi(x)$. For any distribution $\mathcal{D}$ over $(x, y)$, let $Q$ be the induced marginal distribution of $z$. We denote by $Q_t$ the target marginal of $z$ and by $Q_b(w)$ the bridge marginal induced by the mixture weight $w \in \Delta_K$. Let the predictor induced by the composite prompt $P_b(w)$ be denoted by $h_w$ and define the target risk $R_t(w) := \mathbb{E}_{(x,y)\sim\mathcal{D}_t}\big[\ell(h_w(x), y)\big]$.

**Assumption A.3** (Covariate shift and Lipschitz function). There exists a conditional label mechanism $\eta(y \mid z)$ such that for every bridge mixture $w \in \Delta_K$, the joint distributions satisfy

$$(z, y) \sim Q_t(z)\,\eta(y \mid z), \qquad (z, y) \sim Q_b(w)(z)\,\eta(y \mid z).$$

For every $w \in \Delta_K$, define the conditional expected loss

$$g_w(z) := \mathbb{E}_{y\sim\eta(\cdot|z)}\big[\ell(h_w(\phi^{-1}(z)), y)\big].$$

Assume $g_w$ is $L$-Lipschitz w.r.t. $\| \cdot \|_2$, *i.e.*, $|g_w(z) - g_w(z')| \leq L\|z - z'\|_2$ for all $z, z'$. Also assume $\ell \in [0, 1]$.

**Assumption A.4** (Gaussian model). The representation marginals are modeled (or approximated) as Gaussians: $Q_t = \mathcal{N}(\mu_t, \Sigma_t)$ and $Q_b(w) = \mathcal{N}(\mu_b(w), \Sigma_b(w))$, where $(\mu_t, \Sigma_t) = \Gamma_t$ and $(\mu_b(w), \Sigma_b(w)) = \Gamma_b(w)$. The distance $\mathcal{W}(\Gamma_t, \Gamma_b(w))$ in Eq. 10 is defined as the 2-Wasserstein distance between these two Gaussians: $\mathcal{W}(\Gamma_t, \Gamma_b(w)) := W_2\big(\mathcal{N}(\mu_t, \Sigma_t), \mathcal{N}(\mu_b(w), \Sigma_b(w))\big)$.

**Assumption A.5** (Uncertainty as a bound of estimation error). There exists an (unknown) oracle predictor $h^\star$ in the same function space such that each asset-induced predictor $h_j$ (corresponding to $\mathcal{A}_j$) can be written as

$$h_j = h^\star + \epsilon_j,$$

where $\epsilon_j$ is a random error term with $\mathbb{E}[\epsilon_j \mid z] = 0$ for all $z$ and the errors are conditionally uncorrelated: $\mathbb{E}[\epsilon_i(z)\epsilon_j(z) \mid z] = 0$ for $i \neq j$. Moreover, the uncertainty score upper bounds the conditional second moment of the error:

$$\mathbb{E}\big[\epsilon_j(z)^2 \mid z\big] \leq u_j \qquad \text{for all } z \text{ and all } j.$$

Finally, the composite prompt induces the convex combination predictor

$$h_w = \sum_{j=1}^K w_j h_j, \qquad w \in \Delta_K.$$

**Proposition A.6** (Certified upper bound on expected target risk). *Under Assumptions A.3–A.5, for any $w \in \Delta_K$,*

$$R_t(w) \leq R^\star + L \cdot \mathcal{W}(\Gamma_t, \Gamma_b(w)) + \sum_{j=1}^K u_j w_j^2, \tag{28}$$

*where $R^\star := \mathbb{E}_{(x,y)\sim\mathcal{D}_t}\big[\ell(h^\star(x), y)\big]$ is the oracle risk. Consequently, minimizing $\mathcal{W}(\Gamma_t, \Gamma_b(w)) + \lambda \sum_j u_j w_j^2$ (Eq. 10) minimizes the $w$-dependent part of the certified bound in (28) up to a constant rescaling of $\lambda$.*

*Proof.* We split the target risk into an oracle term, a distribution-shift term controlled by optimal transport in representation space, and an estimation-stability term controlled by the uncertainty-weighted $\ell_2$ norm of $w$.

**Reduce risks to the representation marginals.** By Assumption A.3 (representation covariate shift), for any $w \in \Delta_K$,

$$R_t(w) = \mathbb{E}_{z\sim Q_t}[g_w(z)], \qquad R_b(w) := \mathbb{E}_{(x,y)\sim\mathcal{D}_b(w)}[\ell(h_w(x), y)] = \mathbb{E}_{z\sim Q_b(w)}[g_w(z)].$$

**Bound the distribution-shift term via Wasserstein distance.** Since $g_w$ is $L$-Lipschitz (Assumption A.3), the Kantorovich–Rubinstein duality yields

$$\mathbb{E}_{Q_t}[g_w] - \mathbb{E}_{Q_b(w)}[g_w] \leq L\,W_1(Q_t, Q_b(w)).$$

Moreover, on $\mathbb{R}^d$ endowed with the Euclidean metric, $W_1(\cdot, \cdot) \leq W_2(\cdot, \cdot)$, hence

$$R_t(w) \leq R_b(w) + L\,W_2(Q_t, Q_b(w)).$$

By Assumption A.4, $W_2(Q_t, Q_b(w))$ equals the Gaussian 2-Wasserstein distance $\mathcal{W}(\Gamma_t, \Gamma_b(w))$ used in Eq. 10. Therefore,

$$R_t(w) \;\leq\; R_b(w) \;+\; L \cdot \mathcal{W}(\Gamma_t, \Gamma_b(w)). \tag{29}$$

**Bound the bridge risk by oracle risk plus an uncertainty-weighted stability term.** Under Assumption A.5, we have

$$h_w = \sum_{j=1}^{K} w_j(h^\star + \epsilon_j) = h^\star + \epsilon_w, \qquad \text{where} \quad \epsilon_w := \sum_{j=1}^{K} w_j \epsilon_j.$$

Taking conditional second moments and using conditional uncorrelatedness,

$$\mathbb{E}[\epsilon_w(z)^2 \mid z] = \mathbb{E}\Big[\Big(\sum_{j=1}^{K} w_j \epsilon_j(z)\Big)^2 \,\Big|\, z\Big] = \sum_{j=1}^{K} w_j^2 \, \mathbb{E}[\epsilon_j(z)^2 \mid z] \leq \sum_{j=1}^{K} w_j^2 u_j.$$

Now consider the bridge risk difference relative to the oracle. By the definition of $g_w$ and the conditional unbiasedness $\mathbb{E}[\epsilon_w \mid z] = 0$, the excess expected loss incurred by using $h_w$ instead of $h^\star$ is controlled by the conditional second moment. Concretely, for bounded losses in $[0,1]$, one can upper bound the expected excess risk by a constant multiple of the mean-square prediction error; in particular, under the common squared-loss instantiation $\ell(h(x), y) = (h(x) - y)^2$ with bounded outputs, the decomposition is exact:

$$\mathbb{E}_{(x,y)\sim\mathcal{D}_b(w)}[\ell(h_w(x), y)] = \mathbb{E}_{(x,y)\sim\mathcal{D}_b(w)}[\ell(h^\star(x), y)] + \mathbb{E}_{z\sim Q_b(w)}[\epsilon_w(z)^2].$$

Thus,

$$R_b(w) \;\leq\; R^\star \;+\; \mathbb{E}_{z\sim Q_b(w)}[\epsilon_w(z)^2] \;\leq\; R^\star + \sum_{j=1}^{K} u_j w_j^2. \tag{30}$$

**Combine the bounds.** Substituting Eq. (30) into Eq. (29) yields

$$R_t(w) \;\leq\; R^\star \;+\; L \cdot \mathcal{W}(\Gamma_t, \Gamma_b(w)) \;+\; \sum_{j=1}^{K} u_j w_j^2,$$

which is exactly (28). This concludes the proof. $\qquad\square$

### A.4. Lipschitz-Stability of Solution

In this subsection, we now analyze the sensitivity of the optimal mixture weights with respect to perturbations in the target domain statistics. Stability is a crucial property for few-shot adaptation, as estimated statistics from limited data often contain noise. We show that our closed-form solution is Lipschitz continuous, guaranteeing that small variations in the target statistics result in bounded changes in the induced policy.

**Proposition A.7.** *(Perturbation Bound) Let $\hat{b} = b + \varepsilon$ be the target statistics perturbed by noise $\varepsilon \in \mathbb{R}^{2C}$. The deviation in the optimal mixture weights is bounded by:*

$$\left\| w^*(\hat{b}) - w^*(b) \right\|_2 \;\leq\; \left\| \mathcal{H}^{-1} A^\top \right\|_2 \left( 1 + \kappa\left(\mathcal{H}^{-1}\right) \right) \cdot \|\varepsilon\|_2, \tag{31}$$

*where $\kappa\left(\mathcal{H}^{-1}\right) = \frac{\lambda_{\max}(\mathcal{H}^{-1})}{\lambda_{\min}(\mathcal{H}^{-1})}$ denotes the condition number of the matrix $\mathcal{H}^{-1}$.*

*Proof.* Let $\Delta w = w^*(\hat{b}) - w^*(b)$. Recalling Eq. (12), the shift in weights is given by:

$$\Delta w = H^{-1}\left( A^\top \varepsilon - \Delta \nu \mathbf{1} \right), \quad \text{with} \quad \Delta \nu = \frac{\mathbf{1}^\top H^{-1} A^\top \varepsilon}{\mathbf{1}^\top H^{-1} \mathbf{1}}. \tag{32}$$

Let $r = H^{-1} A^\top \varepsilon \in \mathbb{R}^K$ denote the *unconstrained gradient response*. This term represents how the weights would shift if there were no equality constraint. We can rewrite the total weight shift as the unconstrained response minus a projection term along the constraint normal:

$$\Delta w = r - \left( \frac{\mathbf{1}^\top r}{\mathbf{1}^\top \mathcal{H}^{-1} \mathbf{1}} \right) \mathcal{H}^{-1} \mathbf{1}. \tag{33}$$

Applying the triangle inequality yields an upper bound on the norm:

$$\|\Delta w\|_2 \leqslant \|r\|_2 + \left| \frac{\mathbf{1}^\top r}{\mathbf{1}^\top \mathcal{H}^{-1}\mathbf{1}} \right| \|\mathcal{H}^{-1}\mathbf{1}\|_2. \tag{34}$$

We rigorously bound the correction term (the second term) by analyzing its components:

First, for the numerator $|\mathbf{1}^\top r|$, we apply the Cauchy-Schwarz inequality:

$$|\mathbf{1}^\top r| \leqslant \|\mathbf{1}\|_2 \|r\|_2 = \sqrt{K}\|r\|_2. \tag{35}$$

Second, for the vector norm $\|\mathcal{H}^{-1}\mathbf{1}\|_2$, we use the spectral bound:

$$\|\mathcal{H}^{-1}\mathbf{1}\|_2 \leqslant \|\mathcal{H}^{-1}\|_2 \|\mathbf{1}\|_2 = \lambda_{\max}(\mathcal{H}^{-1})\sqrt{K}. \tag{36}$$

Third, for the denominator, we utilize the property of the Rayleigh quotient for the positive definite matrix $\mathcal{H}^{-1}$. Specifically, for any vector $x$, $x^\top \mathcal{H}^{-1} x \geq \lambda_{\min}(\mathcal{H}^{-1})\|x\|_2^2$. Setting $x = \mathbf{1}$, we have:

$$\mathbf{1}^\top \mathcal{H}^{-1}\mathbf{1} \geq \lambda_{\min}(\mathcal{H}^{-1})\|\mathbf{1}\|_2^2 = \lambda_{\min}(\mathcal{H}^{-1})K. \tag{37}$$

Combining these three estimates, the scalar coefficient of the correction term simplifies significantly:

$$\begin{aligned}
\left| \frac{\mathbf{1}^\top r}{\mathbf{1}^\top \mathcal{H}^{-1}\mathbf{1}} \right| \|\mathcal{H}^{-1}\mathbf{1}\|_2 &\leqslant \frac{\sqrt{K}\|r\|_2}{\lambda_{\min}(\mathcal{H}^{-1})K} \cdot \left( \lambda_{\max}(\mathcal{H}^{-1})\sqrt{K} \right) \\
&= \frac{K\lambda_{\max}(\mathcal{H}^{-1})}{K\lambda_{\min}(\mathcal{H}^{-1})}\|r\|_2 \\
&= \kappa(\mathcal{H}^{-1})\|r\|_2,
\end{aligned} \tag{38}$$

where $\kappa(\mathcal{H}^{-1}) = \frac{\lambda_{\max}(\mathcal{H}^{-1})}{\lambda_{\min}(\mathcal{H}^{-1})}$ is the condition number. Substituting this back into the triangle inequality:

$$\|\Delta w\|_2 \leqslant \|r\|_2 + \kappa(\mathcal{H}^{-1})\|r\|_2 = \|r\|_2 \left(1 + \kappa(\mathcal{H}^{-1})\right). \tag{39}$$

Finally, substituting the bound for the unconstrained response $\|r\|_2 \leqslant \|\mathcal{H}^{-1}A^\top\|_2\|\varepsilon\|_2$:

$$\|\Delta w\|_2 \leqslant \|\mathcal{H}^{-1}A^\top\|_2 \left(1 + \kappa(\mathcal{H}^{-1})\right) \|\varepsilon\|_2. \tag{40}$$

This concludes the proof. □

# B. Further Experiments

In this section, we provide further empirical evidence to validate the superiority and practicality of IDEA. First, we investigate the cross-task transferability of our asset library to demonstrate the universality of the captured environmental priors. Next, we benchmark our asset-bridge approach against state-of-the-art feedback-driven methods, highlighting our significant advantage in computational cost and latency. Together, these analyses underscore the potential of IDEA as a lightweight, autonomous solution for deployment in real-world embodied agents.

*Table 6.* Experimental results for asset transfer with DUET on **REVERIE** ↔ **R2R**. We highlight the best result with **bold**.

| Shared Assets | Adaptation | REVERIE → R2R | | | R2R → REVERIE | | |
|:---:|:---:|:---:|:---:|:---:|:---:|:---:|:---:|
| | | NE↓ | SR↑ | SPL↑ | TL↓ | SR↑ | RGSPL↑ |
| ✗ | ✗ | 9.78 | 17.33 | 4.82 | 14.67 | 24.91 | 3.47 |
| ✗ | ✔ | 8.77 | 18.56 | 6.31 | 14.60 | 26.88 | 4.05 |
| ✔ | ✗ | 9.63 | 17.99 | 5.41 | 14.75 | 25.86 | 3.78 |
| ✔ | ✔ | **8.29** | **18.77** | **6.70** | **14.17** | **27.58** | **3.89** |

## B.1. Cross-task Asset Transfer

Expanding on our asset transferability analysis, we empirically evaluate performance across distinct navigation tasks by conducting evaluations on the R2R dataset (*i.e.*, fine-grained instruction following task) with a policy trained on the REVERIE dataset (*i.e.*, high-level goal grounding task), and similarly, on the REVERIE dataset using a policy trained on R2R. Evaluations are conducted on the *Val Unseen* split of both benchmarks.

The results in Tab. 6 demonstrate the robust versatility of our proposed assets, yielding two key observations: 1) Despite the significant semantic gap between high-level instructions (REVERIE) and step-by-step commands (R2R), leveraging shared assets yields immediate performance gains even without online adaptation. For instance, in the *R2R → REVERIE* setting, utilizing shared assets alone improves SR by approximately $1\%$. This suggests that the constructed assets capture functional priors which remain transferable regardless of the specific navigation logic. 2) The combination of shared assets and online adaptation consistently yields the best performance across both tasks. This confirms that cross-task assets can serve as a universal initialization, effectively bridging the domain gap and reducing the optimization burden for the TTA process, even with differences in the source tasks.

## B.2. Comparison with Feedback-driven Methods

**Feedback-driven Adaptation**  Unlike standard TTA methods that operate in a self-training mode, feedback-driven approaches introduce external guidance which are derived from Foundation Models (FMs) or human interaction, to correct model updates and rectify trajectories. Intuitively, the efficacy of these methods heavily relies on the quality of the guidance, necessitating powerful MLLMs (*e.g.*, GPT-4o) or human experts. However, this dependency inevitably incurs prohibitive computational overhead and latency. Specifically, we compare our method against three representative works: **RLCF** (Zhao et al., 2023b), which employs a set of CLIP models to provide vision-language alignment scores as reward signals; **ATENA** (Ko et al., 2025), which adopts an active learning paradigm, soliciting human feedback only when the agent's uncertainty exceeds a threshold; and **FeedTTA** (Kim et al., 2025), which explores replacing the human oracle with GPT-4o to provide binary feedback for reinforcement learning.

**Experimental Results**  We present a detailed comparison between feedback-driven and self-supervised (without feedback) strategies across three benchmarks. The results in Tab. 7, Tab. 8, and Tab. 9, yield the following critical insights:

**1) Comparable Performance without External Costs.** Remarkably, IDEA achieves performance that is competitive with, and sometimes superior to, feedback-driven methods, particularly in unseen splits characterized by significant domain gaps. For instance, on the challenging REVERIE *Test Unseen* split (Table 1), IDEA (with DUET) achieves the highest success rate of $55.12\%$, surpassing all three feedback-driven competitors by margins of up to $+0.8\%$. This validates that our *asset-based bridge* mechanism effectively mines sufficient supervision from historical data itself, thereby eliminating the need for expensive external teachers while maintaining robustness.

*Table 7.* Experimental results for different TTA strategies on REVERIE dataset.

| | Methods+Model | REVERIE Val seen | | | | REVERIE Val unseen | | | | REVERIE Test unseen | | | | Time(ms) |
|---|---|---|---|---|---|---|---|---|---|---|---|---|---|---|
| | | OSR↑ | SR↑ | SPL↑ | RGSPL↑ | OSR↑ | SR↑ | SPL↑ | RGSPL↑ | OSR↑ | SR↑ | SPL↑ | RGSPL↑ | |
| With Feedback | HAMT (Chen et al., 2021) | 47.65 | 43.29 | 40.19 | 25.18 | 36.84 | 32.95 | 30.20 | 17.28 | 33.41 | 30.40 | 26.67 | 13.08 | 74.9 |
| | + RLCF (Zhao et al., 2023b) | 47.92 | 44.21 | 41.13 | 25.97 | 37.37 | 33.11 | 30.23 | 17.24 | 33.11 | 30.21 | 26.13 | 12.97 | $2.32 \times 10^3$ |
| | + FeedTTA (Kim et al., 2025) | 62.97 | 55.80 | 49.70 | 31.80 | 40.73 | 35.05 | 31.60 | 17.83 | 38.62 | 34.14 | 29.07 | 14.36 | $7.32 \times 10^3$ |
| | + ATENA (Ko et al., 2025) | 52.92 | 57.34 | 48.08 | 29.60 | 38.85 | 34.00 | 30.96 | 17.51 | 38.19 | 32.55 | 28.38 | 14.32 | $5.13 \times 10^3$ |
| | DUET (Chen et al., 2022c) | 73.86 | 71.75 | 63.94 | 51.14 | 51.07 | 46.98 | 33.73 | 23.03 | 56.91 | 52.51 | 36.06 | 22.06 | 123.2 |
| | + RLCF (Zhao et al., 2023b) | 74.67 | 73.84 | 64.97 | 52.47 | 53.17 | 49.23 | 34.76 | 23.46 | 57.32 | 52.92 | 36.40 | 22.23 | $2.43 \times 10^3$ |
| | + FeedTTA (Kim et al., 2025) | 86.16 | 84.19 | 75.54 | 60.32 | 71.60 | 66.49 | 45.38 | 30.75 | 58.76 | 53.58 | 37.66 | 24.10 | $7.64 \times 10^3$ |
| | + ATENA (Ko et al., 2025) | 85.52 | 84.33 | 74.31 | 59.99 | 71.88 | 68.11 | 45.82 | 31.26 | 57.74 | 54.28 | 40.70 | 25.01 | $5.42 \times 10^3$ |
| Without Feedback | HAMT (Chen et al., 2021) | 47.65 | 43.29 | 40.19 | 25.18 | 36.84 | 32.95 | 30.20 | 17.28 | 33.41 | 30.40 | 26.67 | 13.08 | 74.9 |
| | + Tent (Wang et al., 2021) | 46.03 | 43.43 | 40.78 | 25.81 | 32.60 | 30.56 | 28.23 | 14.48 | 25.06 | 23.73 | 21.78 | 10.82 | 147.7 |
| | + FSTTA (Gao et al., 2024) | 48.21 | 42.87 | 39.56 | 24.58 | 36.78 | 32.89 | 30.51 | 17.20 | 33.39 | 30.39 | 26.65 | 13.61 | 613.4 |
| | + ReCAP (Hu et al., 2025b) | 48.49 | 44.06 | 40.69 | 25.46 | 37.04 | 33.06 | 30.28 | 17.37 | 34.11 | 30.51 | 24.27 | 13.11 | 213.5 |
| | **+ IDEA (Ours)** | 50.67 | 47.33 | 42.13 | 26.82 | 39.87 | 34.92 | 31.52 | 17.76 | 38.14 | 32.81 | 28.52 | 14.45 | 245.8 |
| | DUET (Chen et al., 2022c) | 73.86 | 71.75 | 63.94 | 51.14 | 51.07 | 46.98 | 33.73 | 23.03 | 56.91 | 52.51 | 36.06 | 22.06 | 123.2 |
| | + Tent (Wang et al., 2021) | 73.72 | 71.89 | 64.06 | 50.41 | 51.43 | 47.55 | 33.99 | 23.32 | 57.12 | 52.61 | 36.17 | 22.16 | 258.9 |
| | + FSTTA (Gao et al., 2024) | 75.59 | 75.48 | 65.84 | 52.23 | 56.26 | 54.15 | 36.41 | 23.56 | 58.44 | 53.40 | 36.43 | 22.40 | 833.6 |
| | + ReCAP (Hu et al., 2025b) | 75.06 | 74.72 | 65.87 | 53.42 | 56.67 | 54.74 | 36.22 | 23.74 | 57.72 | 53.07 | 36.52 | 22.47 | 359.4 |
| | **+ IDEA (Ours)** | 78.45 | 78.24 | 67.74 | 55.07 | 58.51 | 56.92 | 38.03 | 25.47 | 58.91 | 55.12 | 39.84 | 24.52 | 344.3 |

*Table 8.* Experimental results on the R2R dataset.

| Model+Method | R2R Val Seen | | | | R2R Val Unseen | | | |
|---|---|---|---|---|---|---|---|---|
| | TL↓ | NE↓ | SR↑ | SPL↑ | TL↓ | NE↓ | SR↑ | SPL↑ |
| *With Feedback* | | | | | | | | |
| DUET | 12.33 | 2.28 | 79 | 73 | 13.94 | 3.31 | 72 | 60 |
| + RLCF | 12.28 | 2.25 | 79 | 74 | 13.87 | 3.12 | 73 | 61 |
| + FeedTTA | 11.49 | 2.09 | 80 | 75 | 13.52 | 2.95 | 75 | 65 |
| + ATENA | 11.27 | 2.18 | 80 | 75 | 12.31 | 2.90 | 75 | 66 |
| BEVBert | 13.56 | 2.17 | 81 | 74 | 14.55 | 2.81 | 75 | 64 |
| + RLCF | 12.94 | 2.23 | 80 | 74 | 14.11 | 2.76 | 75 | 64 |
| + FEEDTTA | 11.88 | 2.17 | 82 | 77 | 12.24 | 2.77 | 75 | 66 |
| + ATENA | 10.79 | 2.26 | 82 | 78 | 12.22 | 2.78 | 76 | 68 |
| *Feedback* | | | | | | | | |
| DUET | 12.33 | 2.28 | 79 | 73 | 13.94 | 3.31 | 72 | 60 |
| + FSTTA | 13.39 | 2.25 | 79 | 73 | 14.64 | 3.03 | 75 | 62 |
| + ReCAP | 12.02 | 2.27 | 78 | 73 | 13.39 | 3.28 | 72 | 61 |
| + IDEA (Ours) | 11.23 | 2.03 | 81 | 76 | 12.47 | 2.91 | 76 | 67 |
| *Without* | | | | | | | | |
| BEVBert | 13.56 | 2.17 | 81 | 74 | 14.55 | 2.81 | 75 | 64 |
| + FSTTA | 12.28 | 2.31 | 80 | 75 | 13.96 | 2.89 | 74 | 63 |
| + ReCAP | 12.31 | 2.27 | 81 | 74 | 12.94 | 2.78 | 75 | 64 |
| + IDEA (Ours) | 10.89 | 2.15 | 83 | 79 | 12.03 | 2.53 | 76 | 68 |

*Table 9.* Experimental results on the R2R-CE dataset.

| Model+Method | R2R-CE Val Seen | | | | | R2R-CE Val Unseen | | | | |
|---|---|---|---|---|---|---|---|---|---|---|
| | TL↓ | NE↓ | OSR↑ | SR↑ | SPL↑ | TL↓ | NE↓ | OSR↑ | SR↑ | SPL↑ |
| *With Feedback* | | | | | | | | | | |
| BEVBert | 13.98 | 3.77 | 73 | 68 | 60 | 13.27 | 4.57 | 67 | 59 | 50 |
| + RLCF | 13.74 | 3.56 | 74 | 69 | 61 | 13.15 | 4.53 | 66 | 60 | 50 |
| + FEEDTTA | 13.54 | 3.08 | 79 | 73 | 64 | 16.15 | 4.33 | 69 | 61 | 50 |
| + ATENA | 11.31 | 3.24 | 75 | 71 | 64 | 13.48 | 4.50 | 67 | 60 | 51 |
| ETPNav | 11.78 | 3.95 | 72 | 66 | 59 | 11.99 | 4.71 | 65 | 57 | 49 |
| + RLCF | 11.45 | 3.91 | 72 | 66 | 59 | 11.71 | 4.67 | 65 | 57 | 49 |
| + FeedTTA | 10.88 | 3.85 | 72 | 67 | 61 | 11.99 | 4.47 | 66 | 58 | 50 |
| + ATENA | 10.81 | 3.86 | 72 | 67 | 61 | 12.89 | 4.53 | 66 | 58 | 49 |
| *Self-supervised* | | | | | | | | | | |
| BEVBert | 13.98 | 3.77 | 73 | 68 | 60 | 13.27 | 4.57 | 67 | 59 | 50 |
| + FSTTA | 14.07 | 4.11 | 74 | 69 | 60 | 13.11 | 4.39 | 65 | 60 | 51 |
| + ReCAP | 12.40 | 3.31 | 76 | 71 | 63 | 13.01 | 4.57 | 66 | 60 | 50 |
| + IDEA (Ours) | 12.27 | 3.04 | 78 | 73 | 64 | 12.67 | 4.26 | 69 | 62 | 52 |
| ETPNav | 11.78 | 3.95 | 72 | 66 | 59 | 11.99 | 4.71 | 65 | 57 | 49 |
| + FSTTA | 11.35 | 3.93 | 72 | 66 | 59 | 11.57 | 4.77 | 64 | 57 | 49 |
| + ReCAP | 11.31 | 3.92 | 72 | 66 | 60 | 11.56 | 4.74 | 65 | 57 | 50 |
| + IDEA (Ours) | 10.87 | 3.82 | 73 | 68 | 62 | 11.17 | 4.47 | 67 | 59 | 51 |

**2) Orders of Magnitude Faster Inference.** The most significant advantage of IDEA lies in its efficiency, which is critical for real-time deployment. As shown in the **Time(ms)** column of Tab. 7, feedback-driven methods suffer from severe latency due to FM inference or interaction loops. Specifically, FeedTTA requires $7.64 \times 10^3$ ms per episode, and ATENA requires $5.42 \times 10^3$ ms. In sharp contrast, IDEA requires only about 300 ms, making it approximately **20×** faster than FeedTTA and **15×** faster than ATENA. This confirms that IDEA is the viable solution among high-performing methods for time-sensitive navigation tasks.

## C. Additional Ablation Study

In Section 6 of the main paper, we provided a comprehensive analysis of the hyperparameter effects of the prompt length $L$ and the library capacity budget $K_{max}$. In this section, we extend our analysis by further investigating the sensitivity of two other critical hyper-parameters: the regularization strength $\lambda$ and the coverage threshold $\tau$. All experiments are conducted on the REVERIE validation unseen split, providing additional insights into the effectiveness and robustness of our method.

### C.1. Sensitivity of $\lambda$ in IDEA

The coefficient $\lambda$ in Eq. 10 governs the impact of the uncertainty-aware regularizer during bridge construction. As shown in Fig. 6(a), increasing $\lambda$ progressively penalizes the contributions of unreliable assets, ensuring that the bridge is dominated by low-uncertainty priors. This filtering effect leads to improved performance, which peaks at $0.4$. This optimum signifies a balanced trade-off between the uncertainty penalty and the distributional alignment objective. However, when $\lambda$ exceeds $0.8$, the performance begins to degrade. This is likely because over-regularization forces the weight distribution towards sparsity that ignores the actual statistical similarity, hindering the optimal projection. Based on these results, we identify $[0.2, 0.8]$ as the robust operating range where IDEA consistently outperforms the prior SOTA (ReCAP) and select $\lambda = 0.4$ by default.

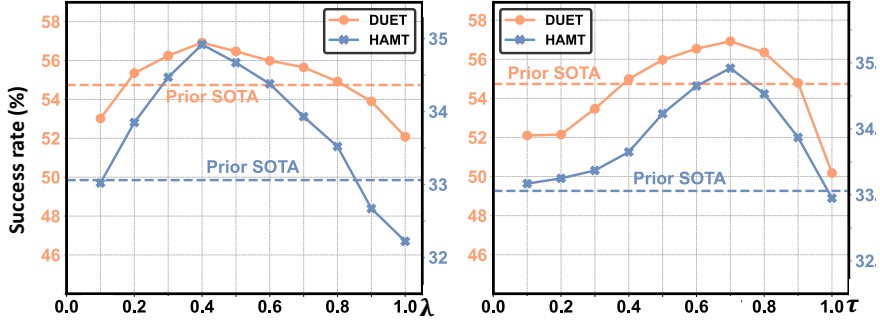

*Figure 6.* Sensitivity analysis on the REVERIE val unseen split: (a) Uncertainty regularization strength $\lambda$ and (b) Coverage threshold $\tau$.

## C.2. Sensitivity of $\tau$ in IDEA

The parameter $\tau$ acts as a gatekeeper, controlling the trade-off between utilizing the lightweight bridge and triggering the asset optimization process. As illustrated in Fig. 6(b), performance improves as $\tau$ increases, reaching a peak Success Rate (SR) at 0.7. This indicates that a moderately relaxed threshold allows the agent to effectively leverage the bridge for reducing distribution shifts, accelerating adaptation without incurring optimization costs. However, when $\tau$ exceeds the optimal range $[0.1, 0.8]$, performance begins to degrade significantly. This suggests that an overly permissive threshold accepts suboptimal bridges that fail to sufficiently reduce the domain gap, thereby introducing noisy priors and leading to negative transfer. Based on this observation, we set $\tau = 0.7$ as the default, balancing bridge utilization efficiency with rigorous quality control.

# D. More Implementation Details

## D.1. Additional Details of Statistic Computing and Alignment

In this subsection, we provide the explicit mathematical formulations for computing the feature statistics and the layer-wise alignment objective used in our asset optimization.

**Feature Statistics Computation.** As described in the main paper, at each navigation step $t$, the model processes an exploration graph $G_t$, containing $N$ navigable candidate nodes.

Let $\mathcal{Z}_t^\ell(P) \in \mathbb{R}^{N \times C}$ denote the set of multi-modal tokens obtained from the $\ell$-th transformer layer with prompt injection, where $z_{t,i}^\ell \in \mathbb{R}^C$ represents the feature vector for the $i$-th candidate node.

We compute the first-order (mean) and second-order (standard deviation) statistics by pooling over the candidate node dimension $N$. Formally, the mean vector $\mu_t^{(\ell)} \in \mathbb{R}^C$ and the standard deviation vector $\sigma_t^{(\ell)} \in \mathbb{R}^C$ for layer $\ell$ are calculated as follows:

$$\mu_t^{(\ell)} = \frac{1}{N} \sum_{i=1}^N z_{t,i}^{(\ell)}, \quad \sigma_t^{(\ell)} = \sqrt{\frac{1}{N-1} \sum_{i=1}^N \left(z_{t,i}^{(\ell)} - \mu_t^{(\ell)}\right)^2 + \epsilon}, \tag{41}$$

where the squaring operation and the square root are applied element-wise, and $\epsilon = 1e^{-6}$ is a small constant for numerical stability. These statistics $\Gamma_t^{(\ell)} = \left\{\mu_t^{(\ell)}, \sigma_t^{(\ell)}\right\}$ serve as a compact descriptor of the current visual-linguistic context, capturing the distributional properties of the navigable space.

**Source Statistics Precomputation.** The source domain statistics $\Gamma_S^{(\ell)} = \left\{\mu_S^{(\ell)}, \sigma_S^{(\ell)}\right\}$ are precomputed offline. We randomly sample a subset of 128 trajectories from the training data. For each trajectory, we collect the feature tokens from all steps and compute the global mean and standard deviation across all sampled nodes. These fixed source statistics serve as the anchor for our online moment-matching objective. Since we only store compact statistical vectors (dimension $C = 768$, the number of fusion layers $M = 4$) rather than raw data, the total storage including both the precomputed source statistics and the dynamic asset library, is merely **0.58 MB**. This negligible overhead corresponds to less than **0.1**% of the source model size, making our framework exceptionally lightweight and suitable for resource-constrained deployment.

**Layer-wise Alignment.** The alignment loss aims to minimize the discrepancy between the current online statistics and the source anchors. Substituting the computed definitions into the objective function (Eq. 5 in the main paper), the explicit optimization problem for prompt $P$ becomes:

$$\min_P \sum_{\ell=1}^M \alpha_\ell \left( \|\mu_S^{(\ell)} - \mu_t^{(\ell)}(P)\|_2 + \|\sigma_S^{(\ell)} - \sigma_t^{(\ell)}(P)\|_2 \right). \tag{42}$$

For the initial asset (*i.e.*, the first novel domain encountered), we initialize the prompt using a random Gaussian distribution. For all subsequent assets, we employ the constructed bridge prompt as a warm-start initialization. Each asset is optimized for $O = 50$ steps.

---

**Algorithm 1:** **I**nter-**D**omain Bridg**E** with Historical **A**ssets (IDEA)

---

**Input:** Task stream $\mathcal{X} = \{X_i\}_{i=1}^N$ (sequence of navigation episodes), pre-trained policy $\pi_\theta$, initial asset library $\mathcal{M} = \emptyset$, capacity budget $K_{\max}$, optimization steps $O$, threshold $\tau$, regularization $\lambda$.

**Output:** Action sequence for each task.

1 **for** *Task $X_i \in \mathcal{X}$* **do**
2     Initialize agent state $s_0$ and step $t = 0$;
3     **while** *not stop* **do**
4        Obtain observation $s_t$ and extract statistics $\Gamma_t = \{\mu_t, \sigma_t\}$;
         `// 1.  Try Bridge Construction (Shortcut)`
5        Compute mixture weights $w$ over $\mathcal{M}$ via closed-form solution ;             `// (Eq. 12)`
6        Construct bridge prompt $P_b = \sum w_k P_k$ and induced statistics $\Gamma_b$;;          `// (Eq. 9)`
7        Calculate discrepancy: $d_p = \mathcal{W}(\Gamma_t, \Gamma_S)$ vs. $d_0 = \mathcal{W}(\Gamma_b, \Gamma_S)$;
8        **if** $d_p < \tau \cdot d_0$ *(**Covered**)* **then**
9           Set current prompt $P_t^* \leftarrow P_b$;
10        **else**
11           Initialize $P$ (warm-start with $P_b$); Compute Fisher-guided Weighting $\alpha$ based on current sensitivity;
            `// (Eq. 8)`
12           Optimize the prompt $P$ in multi-layer alignment for $O$ steps;          `// (Eq. 5)`
13           Set current prompt $P_t^* \leftarrow P$;
14           Form a new asset $\mathcal{A}^* = \{P_t^*, \Gamma_t, \mathrm{Ent}(\pi_{\theta, P_t^*})\}$;
15           **if** $|\mathcal{M}| < K_{\max}$ **then**
16              $\mathcal{M} \leftarrow \mathcal{M} \cup \{\mathcal{A}^*\}$;
17           **else**
18              Find nearest neighbor: $k = \arg\min_j d(\Gamma_j, \Gamma_t)$; Merge asset: $\mathcal{A}_k \leftarrow \frac{1}{2}(\mathcal{A}_k + \mathcal{A}^*)$;
19           **end**
20        **end**
21        Predict and execute action $a_t \sim \pi_{\theta, P_t^*}(s_t)$; Transition to next state $s_{t+1}$, $t \leftarrow t + 1$;
22     **end**
23 **end**

---

## D.2. Pseudocode of our IDEA

Algorithm 1 outlines the complete execution workflow of our proposed framework. Operating at the step level, IDEA continuously monitors the domain shift for each navigation step. The procedure follows a conditional dual-branch logic: 1) It first attempts to construct a training-free bridge by compositing historical assets. If the coverage criterion is met, the bridge is directly deployed for efficient inference. 2) Otherwise, it triggers the online optimization process to acquire a new asset, which is subsequently merged into the library to incrementally expand the agent's knowledge base.

## E. More Experimental Details

### E.1. More Details on Dataset

In this paper, we primarily evaluate the robustness and adaptability of all methods using the widely adopted benchmark: REVERIE (**R**emote **E**mbodied **V**isual r**E**ferring exp**R**ession **I**n **R**eal-**I**ndoor **E**nvironments) (Qi et al., 2020). Derived from the Matterport3D simulator (Chang et al., 2017), REVERIE comprises 10,567 panoramic images and 21,702 high-level instructions spanning 90 distinct buildings. The benchmark is designed to assess an agent's ability to localize remote objects described by natural language. Specifically, it features concise instructions targeting specific items (*e.g.*, *"Find the cushion on the sofa in the living room"*), requiring the agent to simultaneously perform long-horizon navigation and fine-grained object grounding. The dataset is partitioned into *Val Seen*, *Val Unseen*, and *Test Unseen* splits. Crucially, the *Val/Test Unseen* splits involve environments strictly disjoint from the training set, simulating real-world deployment scenarios where agents

must generalize to novel architectural layouts and unseen object appearances.

Additionally, we conduct experiments on two other challenging benchmarks, R2R (Anderson et al., 2018) and R2R-CE (Krantz et al., 2020), to further validate the versatility of our method across different navigation tasks. **R2R** (Room-to-Room) is built upon discrete connectivity graphs, comprising 10,800 panoramic views and 7,189 trajectories. It focuses on fine-grained instruction following, where the agent must strictly adhere to detailed step-by-step commands (*e.g.*, *"Walk past the kitchen, turn left at the hallway..."*). Consequently, this dataset places a premium on trajectory fidelity and precise local scene understanding. **R2R-CE** (Continuous Environment) elevates the challenge by deploying agents in continuous 3D spaces rather than pre-defined graphs. Containing 16,000 instruction-trajectory pairs, it introduces realistic low-level control noise and obstacle collision dynamics. This setting demands robust adaptability to continuous sensor streams and complex physical interactions, testing the agent's stability beyond discrete decision-making.

### E.2. Details of Evaluation Metrics

We employ standard metrics for Vision-Language Navigation (VLN) to comprehensively assess agent performance. The details are as follows:

**Trajectory Length.** TL measures the average total path length traversed by the agent, expressed in meters. While not a direct indicator of success, a lower TL (closer to the shortest path length) generally implies more efficient exploration without redundant movements.

**Navigation Error.** NE is defined as the average geodesic distance (in meters) between the agent's final predicted location and the ground-truth target location. A lower NE indicates higher precision in locating the destination.

**Success Rate.** SR is the primary metric for navigation accuracy. It represents the percentage of episodes where the agent successfully stops within a threshold distance (typically 3 meters) of the target location. Formally, $SR = \frac{1}{N} \sum_{i=1}^{N} \mathbb{I}[NE_i < 3m]$, where $N$ is the total number of episodes.

**Oracle Success Rate.** OSR measures the potential for success. It calculates the percentage of episodes where *at least one point* along the agent's traversed trajectory falls within the success threshold (3m) of the target, regardless of where the agent actually stopped. A large gap between SR and OSR typically indicates a failure in stop condition recognition.

**Success penalized by Path Length.** SPL evaluates the weighed trajectory efficiency for navigation success, where a score closer to SR indicates that the trajectory closely followed the shortest path. The metric is defined as: $SPL = \frac{1}{N} \sum_{n=1}^{N} S_n \frac{TL_n}{\max(SP_n, TL_n)}$, where $S$ is the binary indicator for success and SP denotes the shortest path. Higher SPL indicates highly efficient and accurate navigation.

**Remote Grounding Success penalized by Path Length.** RGSPL extends the SPL concept to the object grounding task. It measures the efficiency of successfully identifying the target object (Remote Grounding Success) weighted by the path length taken to reach the viewpoint from which the object is identified.

