# OpenReview forum: "Turning Adaptation into Assets: Cross-Domain Bridging for Online Vision-Language Navigation"
_ICML.cc/2026/Conference — ICML 2026 regular_

### Official Review · Reviewer_FxZL · 2026-03-08

**Soundness:** 3
**Presentation:** 3
**Significance:** 2
**Originality:** 2
**Overall Recommendation:** 4
**Confidence:** 3

**Summary:**

This paper proposes IDEA, a test-time adaptation framework for Vision-Language Navigation (VLN) that treats adaptation knowledge as reusable assets instead of transient updates. It maintains a historical asset library of compact prompts representing past adaptation knowledge and retrieves relevant assets for new environments. A Fisher-guided weighting scheme improves transferability, while a training-free cross-domain bridge combines historical assets to initialize the agent without iterative optimization.

**Compliance With Llm Reviewing Policy:**

Affirmed.

**Final Justification:**

Given the overall quality of the manuscript and the rebuttal, I am keeping my positive score.

**Key Questions For Authors:**

Please refer to the weakness section and solve the issues there.

**Limitations:**

Yes.

**Strengths And Weaknesses:**

${\bf Strengths:}$

+This paper introduces the idea of treating adaptation knowledge as reusable assets rather than ephemeral updates.

+The proposed cross-domain bridge avoids iterative gradient updates during adaptation by computing a closed-form solution.

+Experiments on R2R, REVERIE, and R2R-CE demonstrate consistent improvements over existing methods.

${\bf Weaknesses:}$

-Experiments focus only on single-agent navigation scenarios. It remains unclear how the asset library would scale to multi-agent systems or collaborative environments.

-Assets are coupled with the feature representation of the underlying model. This may limit transferability across architectures or backbone encoders without additional adaptation.

-This paper does not deeply analyze how asset selection behaves under large domain shifts or noisy environments. More ablations on asset library size and retrieval criteria would strengthen the claims.

-As the agent navigates more environments, the library of "Historical Assets" could grow significantly. The paper lacks a detailed analysis of the memory overhead and the potential computational latency introduced by searching through a large asset bank during real-time navigation.

-The effectiveness of the Cross-Domain Bridge relies heavily on the quality of the soft prompts and the distribution matching process. In environments with extremely high visual noise or ambiguous instructions, the retrieval mechanism might select irrelevant assets, leading to negative transfer.

-While the paper tests on standard benchmarks, the types of "non-stationary shifts" are largely restricted to indoor household settings. It remains unclear how well the asset-based composition would scale to more radical domain shifts, such as moving from indoor to complex outdoor urban environments.

---

> ### Author Rebuttal · Authors · 2026-03-30
>
> We deeply appreciate your positive feedback and constructive suggestions on improving our paper. Building on your comments, we were able to further demonstrate the robustness and effectiveness of IDEA by incorporating additional explanations and experiments.
>
> > W1: It remains unclear how the asset library would scale to multi-agent systems or collaborative environments.
>
> A1: Thank you for this insightful question. IDEA can scale to multi-agent settings in two forms. 1) Offline sharing: an agent that has already deployed IDEA can transfer its stored assets to other agents. This is supported by the asset-transfer results in Tab. 4&6 of the main paper, where reused assets bring consistent gains without rebuilding the library from scratch. 2) Online sharing: multiple agents can collaboratively maintain a shared asset library. As shown in Tab. 6 (response to Reviewer ofiJ), shared IDEA consistently outperforms independent deployment, and performance further improves as the shared library grows.
>
> ---
>
> > W2: Assets are coupled with the feature representation of the underlying model, which may limit transferability across architectures or backbone encoders.
>
> A2: Thank you for this helpful comment. We agree that direct asset transfer across different architectures is not fully addressed in the current design and may require additional alignment or adaptation. Our contribution in this paper is to demonstrate that adaptation knowledge can already be stored and reused effectively across domains (Tab. 4 in the main paper), corrupted scenes (Tab. 11 below), tasks (Tab. 6 in the main paper), and collaborative agents (Tab. 6 in response to Reviewer ofiJ) within a compatible feature space. We will clarify this scope and limitation explicitly in the revision.
>
> ---
>
> > W3: Under large domain shifts or noisy environments, more ablations on asset library size and retrieval criteria would strengthen the claims.
>
> A3: We agree that this analysis would strengthen the paper. We therefore conduct additional ablations under noisy environments by injecting Gaussian noise into the observations. As shown in Tab. 9, the same trends hold as in clean scenes: the proposed bridge consistently outperforms nearest-neighbor retrieval, and larger asset libraries further improve performance.
>
> This provides additional evidence that IDEA is robust to noisy environments and that its storage and retrieval mechanism remains stable under more challenging conditions.
>
> **Table 9: Ablation studies under noisy environments.**
>
> |Model+Method|SR|SPL|
> |:-|:-|:-|
> |DUET|41.22|29.57|
> |+Nearest Retrieval|45.35|32.40|
> |+IDEA Size=32|49.37|34.66|
> |+IDEA Size=48|49.62|34.83|
> |+IDEA Size=64|49.77|34.91|
>
> ---
>
> > W4: As the library grows across more environments, what about the memory overhead and computational latency introduced by searching?
>
> A4: We further analyze memory overhead and search latency as the asset library grows. Storage increases linearly with the maximum library size, while the added search cost remains very small due to the efficient closed-form bridge computation. As shown in Tab. 10, even at library size 128 (i.e., 4× default), the storage overhead is only 2.27 MB, about 0.3% of the source model size. The additional search cost is only 2.1 ms per episode, i.e., less than 1% of total inference time. These results suggest that IDEA scales efficiently even with a substantially larger asset library.
>
> **Table 10: Memory overhead and search latency under different library sizes.**
>
> |Library Size|Mem (MB)|Search Time per Episode (ms)|
> |:-|:-|:-|
> |32|0.58|0.4|
> |64|1.15|0.9|
> |128|2.27|2.1|
>
> ---
>
> > W5: In environments with extremely high visual noise or ambiguous instructions, the retrieval mechanism might select irrelevant assets, leading to negative transfer.
>
> A5: Thank you for raising this important question. We evaluate IDEA under two corrupted settings: visual corruption (Gaussian noise) and text corruption (random token masking to create more ambiguous instructions). As shown in Tab. 11, IDEA consistently outperforms the prior SOTA in both cases, suggesting that its retrieval and bridge construction remain robust under noisy observations and ambiguous language.
>
> **Table 11: Results under visual corruption (VC) and text corruption (TC).**
>
> |Method|VC-SR|VC-SPL|TC-SR|TC-SPL|
> |:-|:-|:-|:-|:-|
> |DUET|41.22|29.57|36.05|26.55|
> |ReCAP|47.48|32.13|39.86|28.93|
> |IDEA|49.37|34.66|40.76|30.21|
>
> ---
>
> > W6: The current benchmarks are still indoor. How well would IDEA scale to more radical shifts, such as indoor → outdoor?
>
> A6: We agree that indoor-to-outdoor transfer is an important and challenging setting. However, it remains underexplored in the current TTA literature and lacks standard benchmarks. Following prior work, we evaluate IDEA on established indoor benchmarks, while further testing its robustness under cross-task and corruption shifts. We will clarify indoor-to-outdoor transfer as an important direction for future work in the revision.

---

> > ### Author Rebuttal · Reviewer_FxZL · 2026-04-01
> >
> > The rebuttal has resolved my primary concerns. However, Table 10 (in rebuttal) would be more informative if it also included a performance comparison against the searching size. Given the overall quality of the revised manuscript, I am keeping my positive score.

---

> > > ### Author Response · Authors · 2026-04-04
> > >
> > > We thank the reviewer for the constructive follow-up and are encouraged that the main concerns have been largely resolved. In response to the suggestion on Table 10, we further provide a detailed comparison below, which jointly reports library size, memory overhead, search latency, and performance under three settings.
> > >
> > > As shown in Tab. 12, increasing the library size **consistently improves performance** in **cross-domain transfer**, **noisy environments**, and **multi-agent collaboration**, while introducing only **modest memory and search overhead**. This further supports that IDEA scales favorably as the asset library grows.
> > >
> > > **Table 12. Effect of library size on memory, search latency, and performance across three settings.**
> > >
> > > | Library Size | Mem (MB) | Search Time / Episode (ms) | REVERIE Val Unseen (SR / SPL) | Visual Noise (SR / SPL) | Multi-Agent (SR / SPL) |
> > > |:--|:--:|:--:|:--:|:--:|:--:|
> > > | 32 (default) | 0.58 | 0.4 | 56.92 / 38.03 | 49.37 / 34.66 | 57.21 / 38.43 |
> > > | 64 | 1.15 | 0.9 | 57.31 / 38.11 | 49.77 / 34.91 | 57.53 / 38.77 |
> > > | 128 | 2.27 | 2.1 | 57.42 / 38.18 | 49.95 / 35.07 | 57.70 / 38.92 |
> > >
> > >
> > > Again, we would like to thank you for appreciating our work and recognizing our contributions!
> > >
> > > Best regards, Authors

---

### Official Review · Reviewer_rYCv · 2026-03-11

**Soundness:** 3
**Presentation:** 3
**Significance:** 3
**Originality:** 3
**Overall Recommendation:** 4
**Confidence:** 3

**Summary:**

This paper proposes the Inter-Domain Bridge with Historical Assets (IDEA) for the test-time adaptation problem in the vln task. Unlike existing TTA methods that treat adaptation as an independent update, IDEA learns transferable knowledge through the design of soft prompts and stores it as assets, thereby addressing the challenges of catastrophic forgetting and negative transfer.

**Compliance With Llm Reviewing Policy:**

Affirmed.

**Key Questions For Authors:**

Please refer to "Weaknesses".

**Limitations:**

Yes.

**Strengths And Weaknesses:**

## Strengths
1. This paper pioneers a knowledge-as-asset paradigm for VLN TTA, the first to transform adaptive knowledge into structured, reusable assets.
2. Experiments on three benchmarks confirm the effectiveness of the proposed IDEA.

## Weaknesses
1. Where do the 128 samples in L176 come from? Are they randomly sampled from the training set?
2. The VLA framework has witnessed rapid advancements in recent times. The authors are advised to validate the efficacy of IDEA on these MLLM-based VLN models (e.g., NaviLLM, StreamVLN, NaVid).

---

> ### Author Rebuttal · Authors · 2026-03-30
>
> We thank the reviewer for the careful review and valuable feedback. We are especially encouraged that the reviewer recognized the novelty of the knowledge-as-asset paradigm and the strong empirical performance on three benchmarks. We will address your questions below.
>
> > W1: Where do the 128 samples in L176 come from? Are they randomly sampled from the training set?
>
> A1: Yes, the 128 samples are randomly sampled from the training set and used to estimate the source-domain statistics in Eq. (4). We will clarify this explicitly in the revision.
>
> We further evaluated robustness to this sampling by running 5 independent trials and reporting mean ± std in Tab. 7. Compared with ReCAP, which also relies on source-domain statistics, IDEA consistently achieves better performance with substantially smaller variance across both HAMT and DUET. This suggests that IDEA is more stable and less sensitive to the particular choice of sampled source data.
>
>
> **Table 7. Repeat runs on REVERIE. Values denote mean (std).**
>
> |Model+Method|V-SR|V-SPL|T-SR|T-SPL|
> |:-|:-|:-|:-|:-|
> |HAMT|32.95|30.20|30.40|26.67|
> |+ReCAP|32.72(1.02)|30.61(0.81)|30.12(0.95)|24.66(0.83)|
> |+IDEA|35.13(0.38)|31.34(0.25)|33.02(0.33)|28.31(0.24)|
> |DUET|46.98|33.73|52.51|36.06|
> |+ReCAP|55.12(0.94)|35.75(0.89)|53.49(1.21)|36.10(0.96)|
> |+IDEA|56.81(0.28)|38.19(0.32)|55.28(0.37)|39.71(0.34)|
>
> ---
>
> > W2: The authors are advised to validate the efficacy of IDEA on the MLLM-based VLN models (e.g., NaviLLM, StreamVLN, NaVid).
>
> A2: Thank you for this valuable suggestion. We further evaluate IDEA on the MLLM-based VLN model NaVid on R2R-CE. As shown in Tab. 8, existing TTA methods such as FSTTA and ReCAP both lead to negative transfer in this setting. In contrast, IDEA keeps the policy network frozen and shifts adaptation into reusable prompt assets, which allows it to consistently improve over the NaVid baseline.
>
> These results validate that IDEA has the potential to generalize beyond the backbones studied in the main paper and remain effective on MLLM-based VLN models. We will include this discussion in the revision.
>
> **Table 8. Experiments of NaVid on R2R-CE Val Unseen.**
>
> |Model+Method|SR|SPL|
> |:-|:-|:-|
> |NaVid|37|36|
> |+FSTTA|35|34|
> |+ReCAP|35|35|
> |+IDEA|**38**|**37**|

---

> > ### Author Rebuttal · Reviewer_rYCv · 2026-04-03
> >
> > Thanks to the authors for providing the supplementary experimental results. So I decide to keep the positive rating.

---

> > > ### Author Response · Authors · 2026-04-04
> > >
> > > Thank you for your positive acknowledgement. We sincerely appreciate your careful review and your recognition that this work **pioneers a knowledge-as-asset paradigm for VLN TTA**. Your support is a great encouragement to us.
> > >
> > > We are glad that our rebuttal has addressed your concerns. Following your suggestion, we have supplemented the rebuttal with additional experiments on the generalizability and robustness of IDEA, especially the **encouraging results on MLLM-based VLN models**.  We will incorporate these new results and the corresponding discussion into the revised paper. If you feel that the rebuttal further strengthens the work, we would greatly appreciate any reconsideration of the overall assessment.
> > >
> > > Thank you again for your time and consideration.
> > >
> > > Best regards, Authors

---

### Official Review · Reviewer_ofiJ · 2026-03-11

**Soundness:** 3
**Presentation:** 3
**Significance:** 3
**Originality:** 4
**Overall Recommendation:** 4
**Confidence:** 4

**Summary:**

This paper introduces IDEA, a novel TTA framework that transforms adaptation into reusable knowledge assets. IDEA builds a dynamic asset library by distilling transferable knowledge into soft prompts through Fisher-guided tuning. It then constructs a training-free adaptation bridge by projecting the target domain onto the convex hull of accumulated historical assets, with optimal mixture weights derived via a closed-form solution. Extensive experiments on multiple VLN benchmarks demonstrate that IDEA consistently outperforms state-of-the-art TTA methods in both performance and efficiency, validating its ability to enable effective and context-aware knowledge reuse.

**Compliance With Llm Reviewing Policy:**

Affirmed.

**Final Justification:**

Thanks to the authors. My concerns have been fully solved. So I decide to keep the positive rating.

**Key Questions For Authors:**

1. The bridge construction relies on minimizing the Wasserstein distance of feature statistics (Section 4.2). Could the authors provide a more intuitive explanation connecting this statistical alignment directly to improved navigation metrics?
2. Library Maintenance & Robustness: Regarding the nearest-neighbor merging strategy used when the library reaches capacity, how does the system handle potential redundancy or conflicting assets? Are there mechanisms to detect and prune poisoned or outdated assets to ensure the library's quality during open-ended online adaptation?
3. Transferability vs. Architecture Coupling: The paper demonstrates asset transferability across domains, yet the assets appear tied to the specific feature dimensions of the backbone. Is the observed transferability primarily driven by domain-invariant semantic priors or specific architectural similarities?

**Limitations:**

yes

**Strengths And Weaknesses:**

Strengths:

1. Solid theoretical foundation: The work addresses catastrophic forgetting via reusable asset accumulation. The methodology, utilizing Fisher-guided prompt tuning and Wasserstein-based bridging with KKT solutions, is theoretically rigorous and clearly proven.
2. Comprehensive experimental design: Extensive evaluations on discrete and continuous benchmarks show the method consistently outperforms TTA approaches on SR and SPL. Ablations validate key components, and low inference time supports lightweight deployment.
3. Innovative Framework: The study pioneers a "plug-and-play" knowledge reuse paradigm for VLN. This novel perspective on accumulating historical experience significantly advances lifelong learning and adaptation on resource-constrained devices.

Weaknesses:

1. The asset library (Section 4.1) is deeply coupled with the specific feature space of the backbone model. This restricts the framework's universality, as assets lack true "plug-and-play" portability across agents with different model architectures or dimensions.
2. While the paper claims potential for "asset sharing," the evaluation is strictly confined to single-agent benchmarks (Section 5). The efficacy of the IDEA framework in multi-agent collaborative scenarios and the impact of library growth on retrieval remain unexplored.

---

> ### Author Rebuttal · Authors · 2026-03-30
>
> We sincerely thank the reviewer for the careful review and positive feedback. We are encouraged that the reviewer recognizes that IDEA significantly advances lifelong learning and adaptation on resource-constrained devices. Motivated by the comments, we further demonstrate the effectiveness and robustness of IDEA with additional explanations and experiments.
>
> > Q1: Could the authors provide a more intuitive explanation connecting Wasserstein alignment of feature statistics to improved navigation metrics?
>
> A1: Thank you for this valuable question. In VLN, navigation performance often degrades when the policy receives features that are misaligned with the representation space learned during training. IDEA mitigates this mismatch through Wasserstein alignment, constructing a bridge toward previously adapted feature spaces and thereby improving performance.
>
> We further verify this empirically: across different scenes, the Wasserstein distance is strongly negatively correlated with performance (Pearson = -0.83 / -0.78 for SR / SPL), showing that better Wasserstein alignment is consistently associated with improved navigation metrics.
>
> **Table 4: W-distance and navigation metrics across different scenes on REVERIE.**
>
> |Scene ID|1|2|3|4|5|6|7|8|9|10|
> |:-|:-|:-|:-|:-|:-|:-|:-|:-|:-|:-|
> |SR|55.1|71.5|31.5|48.8|49.3|35.4|54.0|57.6|69.7|67.9|
> |SPL|40.0|58.3|18.0|34.1|23.5|21.0|26.9|37.6|46.4|53.4|
> |W-dist|6.5|4.9|10.1|7.1|10.2|9.2|5.7|7.2|4.5|6.7|
>
> ---
>
> > Q2: Are there mechanisms to handle redundancy, conflicts, or outdated/poisoned assets?
>
> A2: Yes. IDEA addresses these issues at both the merge and reuse stages. Redundancy / outdated assets: Nearest-neighbor merging compresses redundant assets and continuously refreshes nearby older ones with new assets. Conflicts / poisoned assets: During reuse, IDEA avoids brittle hard retrieval and instead forms the bridge by softly combining multiple assets, making it more robust to partial conflicts. In addition, uncertainty-aware regularization downweights unreliable (e.g., poisoned or low-quality) assets, reducing their influence during bridge construction.
>
> We further validated this by adding explicit modules for redundancy pruning, conflict suppression, outdated-asset removal, and entropy-based poison filtering. Tab. 5 shows that these modules bring only marginal changes (around 0.1 on most metrics), indicating that the base IDEA design is already robust to these cases in practice.
>
> **Table 5. IDEA with explicit detector/pruning modules.**
>
> |Method|V-SR|V-SPL|T-SR|T-SPL|
> |:-|:-|:-|:-|:-|
> |IDEA|56.92|38.03|55.12|39.84|
> |+Redundancy Prune|56.98|37.99|55.18|39.80|
> |+Conflict Suppress|56.82|38.07|55.06|39.70|
> |+Outdated Remove|56.86|37.97|55.14|39.86|
> |+Poison Filter|57.06|38.05|55.26|39.88|
>
> ---
>
> > Q3 & W1: The assets appear tied to the backbone feature space. Is the observed transferability due to domain-invariant priors or architectural similarity?
>
> A3: Thank you for this question. Since different models may differ in feature dimensions and representation spaces, cross-architecture shared knowledge is difficult to parameterize, and our current asset design has the same limitation. The observed transferability is primarily driven by domain priors captured by the assets, with architectural consistency enabling their direct reuse.
>
> Empirically, IDEA already demonstrates effective knowledge reuse across domains (Tab. 4 in the main paper), corrupted scenes (Tab. 11 in response to Reviewer FxZL),  tasks (Tab. 6 in Appendix), and multi-agent settings (Tab. 6 below). These results suggest that the asset design is not limited to a single domain or task, which we view as an important step toward scalable knowledge reuse. We will clarify that extending IDEA to heterogeneous architectures is an important direction for future work.
>
> ---
>
> > W2: The efficacy of the IDEA framework in multi-agent collaborative scenarios and the impact of library growth remain unexplored.
>
> A4: Thank you for this valuable suggestion. We further evaluated IDEA in a shared-library multi-agent setting and observed two findings (Tab. 6).
> 1) **Asset sharing is effective across agents.** With the same library size, Shared IDEA consistently outperforms independent deployment, improving SR/SPL by +0.3/+0.4 on Val Unseen and +0.3/+0.3 on Test Unseen.
> 2) **Library growth remains beneficial in the shared setting.** As the shared library size increases from 32 to 64 to 128, performance improves steadily on both splits.
> These results indicate that IDEA naturally supports multi-agent collaboration via a shared asset library. We will include this experiment and discussion in the revision.
>
> **Table 6. Multi-agent collaboration on REVERIE.**
>
> |IDEA Mode|Asset|Size|V-SR|V-SPL|T-SR|T-SPL|
> |:-|:-:|:-:|:-:|:-:|:-:|:-:|
> |Independent|Private|32|56.92|38.03|55.12|39.84|
> |Shared|Shared|32|57.21|38.43|55.38|40.17|
> |Shared|Shared|64|57.53|38.77|55.68|40.59|
> |Shared|Shared|128|57.70|38.92|55.95|40.86|

---

> > ### Author Rebuttal · Reviewer_ofiJ · 2026-04-01
> >
> > Thanks to the authors. My concerns have been fully solved. So I decide to keep the positive rating.

---

> > > ### Author Response · Authors · 2026-04-04
> > >
> > > Thank you for your positive acknowledgement. We sincerely appreciate your careful review and your recognition of the **theoretical grounding**, **comprehensive experimental validation**, and **the novelty of the knowledge-reuse paradigm**. Your support is truly encouraging to us.
> > >
> > > We are pleased that our rebuttal has adequately addressed your concerns. Following your suggestion, we have supplemented the rebuttal with **additional experiments on the extensibility and robustness of IDEA**, including **encouraging results on multi-agent systems**. These new results, together with the corresponding discussion, will be incorporated into the revised paper. If you feel that the rebuttal further strengthens the work, we would greatly appreciate any reconsideration of the overall assessment.
> > >
> > > Thank you again for your time and consideration.
> > >
> > > Best regards, Authors

---

### Official Review · Reviewer_njGy · 2026-03-13

**Soundness:** 3
**Presentation:** 3
**Significance:** 3
**Originality:** 3
**Overall Recommendation:** 4
**Confidence:** 3

**Summary:**

In this paper, the authors propose IDEA , a novel TTA framework for vision-and-language navigation. It converts online adaptations into reusable “assets” for generalizing to new domains. Concretely, it learns soft visual prompts via source statistics with Fisher-guided layer weighting and constructs an asset library. Moreover, it constructs training-free adaptation bridges with a closed-form solution. Experiments show consistent gains over recent TTA baselines.

**Compliance With Llm Reviewing Policy:**

Affirmed.

**Final Justification:**

Thank you so much, authors. My concerns have been fully resolved, and I will keep my positive rating.

**Key Questions For Authors:**

1. Why are soft prompts inserted into the visual tokens only without considering language tokens?

2. How well does the multivariate Gaussian approximation hold?

**Limitations:**

Yes

**Strengths And Weaknesses:**

Strengths:

S1: The problem is well-motivated, and the solution is intuitive.

S2: The presentation is logical and easy to follow.

S3: The closed-form solution for the domain bridge is computationally efficient.

Weakness:

W1: The bridge prompt is constructed via a linear combination of historical soft prompts. Given the highly nonlinear nature of the Transformer, these interpolation operations require justifications.

W2: Also, the assumption that the feature statistics follow multivariate Gaussian distributions should be justified: whether practical feature distributions of the transformer follow this. The paper lacks empirical justification for the claim that the features follow Gaussian distributions.

W3: The solution to Problem (11) is not optimal, as the authors derive this closed-form solution using only the equality constraint while ignoring w>=0. Therefore, the solution w can be negative, at least from a mathematical perspective. This contradicts the authors' claim that "This derivation allows IDEA to construct an optimal bridge via simple matrix operations." If no approximation bound is provided, this is a heuristic approach rather than an optimal or an effective approximate solution.

The paper is well-motivated and grounded by theoretical analysis. However, given the limitations above, I lean towards recommending weak acceptance instead of a higher recommendation.

---

> ### Author Rebuttal · Authors · 2026-03-30
>
> Thank you for your careful review and positive assessment. We especially appreciate your recognition of the well-motivated design and practical efficiency of our framework. Building on your comments, we can further strengthen the paper with additional explanation and supplementary experiments. We are truly grateful for your valuable feedback.
>
> > W1: Why is a linear combination of historical prompts reasonable, given the nonlinearity of the Transformer?
>
> A1: Thank you for this insightful question. We agree that the Transformer is highly nonlinear and its representation space is complex. Our motivation, therefore, is not to model the full space, but to examine whether useful target prompts can be captured in a subspace spanned by historical assets, for which a linear bridge is sufficient in practice.
>
> We test this directly by replacing our closed-form bridge with a more expressive MLP that predicts the bridge composition from target statistics. The MLP requires substantial optimization on the test data under the same objective, yet it performs very similarly to our closed-form bridge, with all metric differences within ±0.3 (Tab. 1). Moreover, compared with the Nearest Asset baseline, our bridge yields clear gains (+6.5 SR / +2.5 SPL on Val Unseen). These results show that a principled linear bridge is already sufficiently expressive and effective in practice. We will include the detailed comparison in the revision.
>
> **Table 1. Bridge construction strategies on REVERIE Val unseen and Test unseen.**
>
> |Bridge|Type|V-SR|V-SPL|T-SR|T-SPL|
> |:-|:-|:-|:-|:-|:-|
> |Nearest|Linear / Retrieval|50.47|35.53|54.11|37.92|
> |MLP|Nonlinear / Optimized|56.64|37.82|55.41|39.56|
> |Ours|Linear / Closed-Form|56.92|38.03|55.12|39.84|
>
> ---
>
> > W2 & Q2: How well does the multivariate Gaussian approximation hold?
>
> A2: Thank you for this question. We conduct two validation experiments. 1) We compare a single Gaussian fit against a 5-component GMM and observe small KL divergences (0.11 / 0.09 for HAMT / DUET). 2) Using the Gaussian-based bridge consistently reduces the source-target feature distribution gap, as measured by the Kernel-Density-Estimation distance. These results suggest that a multivariate Gaussian serves as a practical and efficient approximation for bridge construction.
>
> **Table 2. Feature distribution analysis on REVERIE Val Unseen.**
>
> |Model|KL|KDE (w/o bridge)|KDE (w bridge)|Reduc.|
> |:-|:-:|:-:|:-:|:-:|
> |HAMT|0.11|233.62|162.13|30.6%|
> |DUET|0.09|138.98|95.34|31.4%|
>
> ---
>
> > W3: The closed-form solution ignores w≥0, so is the solution really “optimal”?
>
> A3: Sorry for the confusion. We will revise the claim: the closed-form step gives the optimal solution under the equality constraint $\Sigma_j  w_j=1$, without the non-negativity constraint.
>
> We impose $w_j ≥0$ in practice for stability, since interpolation is typically more reliable than extrapolation in feature space [1,2]. Empirically, this affects only a small number of cases: across three datasets, more than 98% of the unconstrained coefficients are already non-negative. For the remaining cases, we apply ReLU and renormalization (described in Appendix A.2) to obtain a valid convex combination. We will move this clarification to the main paper and revise the claim.
>
> ---
>
> > Q1: Why are soft prompts inserted into the visual tokens only without considering language tokens?
>
> A4: Thank you for this valuable question. In VLN, unseen environments introduce highly dynamic visual shifts across different steps, while the instruction remains fixed within each episode. This makes visual adaptation more critical and demands stronger robustness to continuous distribution changes.
>
> We also verify this empirically in Tab. 3. Under the same prompt length, visual-only prompts consistently achieve the best performance across all tested lengths, outperforming both language-only and joint visual-language prompts. This suggests that, under a fixed prompt budget, adapting visual tokens is the most effective choice in our setting. We will clarify this motivation and add the comparison in the revision.
>
> **Table 3. Comparison of prompt placements (SR / SPL).**
>
> |Length|L-only|V+L|V-only|
> |:-|:-:|:-:|:-:|
> |2|49.13 / 34.45|51.93 / 35.09|**53.95 / 36.12**|
> |4|51.25 / 35.04|54.28 / 36.52|**56.92 / 38.03**|
> |8|52.02 / 35.51|57.04 / 38.09|**57.21 / 38.22**|
>
> ---
>
>
> >**References**
> >
> >[1] Manifold Mixup: Better Representations by Interpolating Hidden States, ICML 2019.
> >
> >[2] An Online Learning Approach to Interpolation and Extrapolation in Domain Generalization, AISTATS 2022.

---

> > ### Author Rebuttal · Reviewer_njGy · 2026-04-06
> >
> > Thank the authors for their rebuttal, which answers my questions. I will keep my positive rating.

---

> > > ### Author Response · Authors · 2026-04-07
> > >
> > > Thank you for your positive acknowledgement. We sincerely appreciate your valuable feedback and recognition of the **well-motivated design** and **practical efficiency** of our framework. Your support is a great encouragement to us.
> > >
> > > We are glad that our rebuttal has adequately addressed your concerns. Following your suggestion, we supplemented the rebuttal with additional ablation studies and observational experiments that **provide further validation** for the **bridge formulation**, **Gaussian approximation**, **optimization constraints**, and **prompt placement strategy**. We will incorporate these new results and the corresponding discussion into the revised paper. If you feel that the rebuttal further strengthens the work, we would greatly appreciate any reconsideration of the overall assessment.
> > >
> > > Thank you again for your time and consideration.
> > >
> > > Best regards, Authors

---

### Decision · Program_Chairs · 2026-04-30

**Decision:**

Accept (regular)

**Comment:**

All four reviewers provided Weak Accept recommendations and agreed that this paper presents a technically sound and well-motivated contribution to Test-Time Adaptation (TTA) for vision-and-language navigation by introducing an asset-based paradigm that transforms online adaptation into reusable knowledge.

The proposed IDEA framework, including Fisher-guided soft prompt accumulation and an efficient closed-form cross-domain bridge, is both conceptually novel and practically appealing, achieving consistent empirical gains across multiple VLN benchmarks while maintaining computational efficiency.

During the rebuttal process, the authors satisfactorily addressed most reviewer concerns and clarified key implementation details, which led reviewers to maintain or strengthen their positive assessments.

Therefore, the ACs recommend acceptance.